# STAYING IN THE SWEET SPOT: RESPONSIVE REASONING EVOLUTION VIA CAPABILITY-ADAPTIVE HINT SCAFFOLDING

## ABSTRACT

Reinforcement learning with verifiable rewards (RLVR) has advanced the reasoning capabilities of large language models (LLMs). However, existing RLVR methods often suffer from exploration inefficiency due to mismatches between problem difficulty and model capability: overly difficult problems hinder reasoning path discovery, while overly simple problems offer little learning signal. To address this, we first formalize the effect of problem difficulty by quantifying the relationship between loss descent magnitude and rollout accuracy. Building on this analysis, we propose **SEELE**, a supervision-aided RLVR framework that dynamically adjusts problem difficulty to lie within the high-performance region. SEELE augments each training sample by appending a hint (part of a full solution) for difficulty reduction. Unlike previous hint-based approaches, SEELE deliberately computes the hint length for each individual problem to achieve an optimal difficulty. The optimal hint length is determined via multi-round rollout sampling, where an item response theory model fits accuracy–hint pairs from previous rounds to predict the next-round hint. This instance-level, real-time difficulty adjustment aligns problem difficulty with the evolving model capability, thereby improving exploration efficiency. Experiments show that SEELE outperforms Group Relative Policy Optimization (GRPO) and Supervised Fine-tuning (SFT) by **+10.0** and **+8.4** points, respectively, and exceeds the best prior supervision-aided approach by **+3.8** points on average across six math reasoning benchmarks.

## 1 INTRODUCTION

Recent large language models (LLMs) such as OpenAI-o1 (OpenAI et al., 2024), DeepSeek-R1 (DeepSeek-AI et al., 2025a), and Kimi-K2 (Team et al., 2025) have made remarkable breakthroughs in reasoning ability, benefiting from long chain-of-thought that incorporates self-reflection and revision. This capability can be realized through pure reinforcement learning with verifiable rewards (RLVR), which forgoes memorizing annotated reasoning processes and instead exploits the model's inherent capabilities by strengthening correct exploratory behaviors (Chu et al., 2025). At present, RLVR has become the common practice for building high-performance reasoning models.

However, on-policy exploration inherently constrains the learning efficiency, exhibiting strong data dependency (Gao et al., 2025; Dou et al., 2025; Schmied et al., 2025; Yu et al., 2025; Zhang et al., 2025b; Sun et al., 2025). RLVR is driven by the rewards from extensive online sampled rollouts, which collapse to zero when the training problems are too difficult for LLMs to produce a correct response. Conversely, overly simple problems yield nearly all correct rewards, producing minor advantage value. It remains unclear what problem difficulty maximizes learning efficiency and how to curate such data. Moreover, as recent studies (Gandhi et al., 2025; Yue et al., 2025; Zhao et al., 2025) have found, RLVR merely amplifies existing behaviors rather than fostering novel cognitive capabilities, thereby limiting the achievable performance to that of the base model. Supervised fine-tuning (SFT) (Köpf et al., 2023) is a naive way to improve the ability of LLMs before RL with expert data. However, existing works (Zhang et al., 2025c; Chen et al., 2025) show that directly using SFT-then-RL is not an effective way, which even underperforms pure RL.

Figure 1: Comparison between the direct rollout (blue) and hinted rollout (purple). The hint consists of the first few steps from an annotated solution. Adding a hint can simplify the problem and guide the LLM toward completing correct solutions.

To overcome these limitations, recent works have attempted to integrate SFT into the RL framework, enabling synergistic learning when SFT-then-RL is ineffective. LUFFY (Yan et al., 2025) and SRFT (Fu et al., 2025) incorporate parallel off-policy guidance into the rollout set, allowing simultaneous exploration and imitation. UFT (Liu et al., 2025a), TAPO (Wu et al., 2025), StepHint (Zhang et al., 2025a), Hint-GRPO (Huang et al., 2025a), and Prefix-RFT (Huang et al., 2025b) append an off-policy hint prefix to each problem to reduce exploration difficulty. While these methods allow for a certain degree of problem difficulty modulation, they have a salient shortcoming: off-policy guidance is applied statically and indiscriminately across problems and hint levels, causing most training samples' difficulty to mismatch the model's evolving capability. Consequently, for those challenging problems, mild off-policy guidance does not effectively resolve the low efficiency of on-policy exploration, whereas for those easy problems, excessive off-policy intervention may impede the LLM from developing its own reasoning patterns. This observation raises a critical question:

*What is the appropriate problem difficulty under off-policy guidance, and how can the difficulty be dynamically adjusted in accordance with the model's evolving capability?*

In this paper, we propose **SEELE**: *reSponsive rEasoning Evolution via capabiLity-adaptivE hint scaffolding*, a theoretically-grounded supervision-aided RVLR approach that keeps high learning efficiency throughout the whole training stage via dynamically adjusting the proportion of the off-policy prefixes. SEELE explicitly formalizes that an appropriately difficult problem should yield a rollout accuracy of approximately 50% through a theoretical analysis of the loss descent magnitude. By appending an *instance-specific, dynamically determined* hint after the original problem (Figure 1), SEELE is able to control the problem difficulty within the "sweet spot". Unlike previous hint-based methods (Liu et al., 2025a; Huang et al., 2025b), our approach enables per-instance adaptation and involves the model's real-time feedback as training progresses, thereby achieving a more precise alignment between problem difficulty and model capability. Concretely, we split a rollout sampling stage into several rounds, across which we establish a regression model to predict the accuracy given the hinting rate (proportion of the full solution) under the guidance of item response theory (Chen et al., 2021). At each round, we fit the accuracy prediction model using the feedback from the previous rounds and predict how long the current hint should be for a 50% accuracy. We conduct experiments on six math reasoning benchmarks and three general domain reasoning benchmarks, on which SEELE significantly outperforms previous RLVR methods.

Our contributions can be summarized as:

- We present a theoretical analysis showing that the learning efficiency of RLVR algorithms follows a quadratic negative relationship with rollout accuracy, and reaches its maximum when the accuracy is 50%.

- Guided by our theoretical analysis, we propose a novel capability-adaptive RLVR framework that manipulates the problem difficulty via multi-round rollout sampling and accuracy prediction, maintaining high learning efficiency throughout the entire training process.

- We demonstrate SEELE's superiority over previous RLVR methods on 9 challenging benchmarks, achieving remarkable improvements of **+10.0** points on average compared with GRPO on math reasoning.

## 2 RELATED WORK

**Reinforcement Learning for LLM Reasoning.** Recent work has demonstrated the effectiveness of reinforcement learning (RL) in improving the reasoning capabilities of large language models (LLMs), as exemplified by systems such as DeepSeek-R1 (DeepSeek-AI et al., 2025a), OpenAI-o1 (OpenAI et al., 2024), and Kimi-K2 (Team et al., 2025). These studies show that pure RL with verifiable rewards can drive LLMs to autonomously develop extended chain-of-thought reasoning patterns, incorporating substantial self-reflection and iterative refinement. Among these approaches, GRPO-based methods have emerged as a pivotal paradigm for enhancing reasoning performance. Subsequent studies such as DAPO (Yu et al., 2025), Dr.GRPO (Liu et al., 2025b), GSPO (Zheng et al., 2025) focus on addressing GRPO's optimization limitations by removing length bias, difficulty bias, etc. However, recent works (Yue et al., 2025; Wang et al., 2025) argue that RL approaches are bounded by the capabilities of the base model and do not acquire new reasoning skills, which suggests pure RL may not be the ultimate solution.

**Supervision-Aided Reinforcement Learning.** Supervised fine-tuning (SFT) on high-quality reasoning chains effectively strengthens reasoning ability and is typically applied before the RL stage (OpenAI et al., 2024; Qwen et al., 2025). To overcome efficiency and capacity limits of RL, later studies integrate off-policy supervision into RL as a single process. LUFFY (Yan et al., 2025) enriches RL rollouts with off-policy annotated traces for external guidance, while SRFT (Fu et al., 2025) jointly optimizes SFT and RL losses using an entropy-based weighting scheme. UFT (Liu et al., 2025a), StepHint (Zhang et al., 2025a), and Hint-GRPO (Huang et al., 2025a) append partial solutions (hints) from stronger models to handle difficult problems. Prefix-RFT (Huang et al., 2025b) similarly employs hints but excludes low-entropy hint tokens to avoid over-imitation. TAPO (Wu et al., 2025) instead introduces high-level "thought patterns" to promote external strategy learning. Yet these methods rely on static supervision, limiting exploration and imitation when tasks are mismatched to the model's capability. In contrast, SEELE grounds hint assignment in principled RL optimization and introduces a capability-adaptive mechanism, enabling dynamic and explicit difficulty adaptation.

## 3 PRELIMINARIES

**Reinforcement Learning with Verifiable Rewards (RLVR)** formulates the generation process of an LLM as a Markov Decision Process (MDP), where the state is defined as the concatenation of the prompt $x$ and the tokens generated so far $y_{1:t-1}$, and the action corresponds to selecting the next token $y_t$ from the policy, i.e., $y_t \sim \pi_\theta(\cdot|x, y_{1:t-1})$. The objective of RLVR is to train the policy model to generate outputs that achieve high scores under a rule-based reward function $r(x, y)$. Formally, RLVR optimizes the expected reward over the on-policy rollouts generated from the policy model $\pi_{\theta_{old}}$ at the previous step, and the objective can be written as

$$\mathcal{L}(\theta) = \underbrace{-\mathbb{E}_{x \sim \mathcal{D}, y \sim \pi_\theta(\cdot|x)}\big[A_{\theta_{old}}(x, y)\big]}_{\mathcal{L}_{policy}(\theta)} + \beta \underbrace{\mathbb{E}_{x \sim \mathcal{D}}[\mathbb{D}_{KL}(\pi_{\theta_{ref}}(\cdot|x) \| \pi_\theta(\cdot|x)))]}_{\mathcal{L}_{KL}(\theta)}, \quad (1)$$

where $A_{\theta_{old}}(x, y) = r(x, y) - \mathbb{E}_{y \sim \pi_{\theta_{old}}(\cdot|x)}[r(x, y)]$ denotes the advantage function and the hyper-parameter $\beta$ controls the strength of KL regularization with respect to the reference model $\pi_{ref}$.

**GRPO** offers an elegant advantage function implementation, which has been widely applied in RLVR studies (Shao et al., 2024). Given an input $x$, GRPO samples a group of outputs $\{y_1, y_2, \cdots, y_n\}$ from $\pi_{\theta_{old}}(\cdot|x)$ and uses the group mean to replace the expectation. Recently, Liu et al. (2025b) proposes an improved advantage function in their Dr.GRPO approach as

$$A(x, y_i) = r(x, y_i) - \text{mean}(\{r(x, y_j)|j = 1, 2, \cdots, n\}), \quad (2)$$

where the normalization term is removed to mitigate optimization bias. In the following, our analysis is conducted based on this unnormalized formulation.

## 4 METHODOLOGY

In this section, we present SEELE from three aspects: (1) a theoretical foundation that identifies the optimal problem difficulty in terms of rollout accuracy (§ 4.1); (2) a multi-round sampling framework that decomposes rollout generation into sequential rounds, thereby enabling capability-aware

adaptation of problem difficulty (§ 4.2); (3) a rollout accuracy prediction model based on Item Response Theory, which supports accurate and responsive adjustment of problem difficulty (§ 4.3).

## 4.1 Relationship Between Learning Efficiency and Rollout Accuracy

We begin by formulating a quantitative relationship between reinforcement learning (RL) training efficiency and problem difficulty. The model prediction accuracy is employed as the difficulty measure. Given a policy model $\pi_\theta$ and a binary reward function $r(x, y)$, for a problem $x$, we define the prediction accuracy for $x$ with respect to the policy model $\pi_\theta$ as

$$a_\theta(x) = \mathbb{E}_{y \sim \pi_\theta}[r(x, y)]. \tag{3}$$

Prediction accuracy is the expectation of the rollout accuracy and reflects the difficulty of this training instance relative to the current model capability.

Next, we consider a one-step gradient descent from the last-step policy model $\pi_{\theta_{\text{old}}}$ with the update vector $d$. For analytical convenience, we assume $\theta_{\text{ref}} = \theta_{\text{old}}$. The optimization objective (Eq.(1)) can be reformulated by letting $\theta = \theta_{\text{old}} + d$:

$$\min_d \mathcal{L}(\theta_{\text{old}} + d) = \min_d \left\{ \mathcal{L}_{\text{policy}}(\theta_{\text{old}} + d) + \beta \mathbb{E}_{x \sim \mathcal{D}} \left[ \mathbb{D}_{\text{KL}} \left( \pi_{\theta_{\text{old}}}(\cdot|x) || \pi_{\theta_{\text{old}}+d}(\cdot|x) \right) \right] \right\}. \tag{4}$$

We conclude that the magnitude of loss descent is upper bounded by the quadratic envelope of $a_\theta(x)$:

$$\mathcal{L}(\theta_{\text{old}}) - \mathcal{L}(\theta_{\text{old}} + d) \leq \frac{1}{2\beta} \mathbb{E}_{x \sim \mathcal{D}}[a_{\theta_{\text{old}}(x)}(1 - a_{\theta_{\text{old}}}(x))]. \tag{5}$$

Here, we briefly outline the derivation. Since $\mathcal{L}$ is defined as the sum over independent prompts $x$, we analyze the instance-level loss descent $\mathcal{L}(x; \theta_{\text{old}}) - \mathcal{L}(x; \theta_{\text{old}} + d)$ and then aggregate across all instances. From Eq. (4), for a specific instance $x$, its minimizer $d_x^*$ can be approximated by applying first-order Taylor expansion on $\mathcal{L}_{\text{policy}}$ and second-order Taylor expansion on $\mathcal{L}_{\text{KL}}$ at $\theta_{\text{old}}$:

$$d_x^* \approx \arg\min_d \left\{ \mathcal{L}_{\text{policy}}(x; \theta_{\text{old}}) + \nabla_\theta \mathcal{L}_{\text{policy}}(x; \theta)\big|_{\theta=\theta_{\text{old}}}^T d + \frac{\beta}{2} d^T F(\theta_{\text{old}}) d \right\}, \tag{6}$$

where $F(\theta_{\text{old}})$ is the Fisher Information Matrix. Since $F(\theta_{\text{old}})$ is always positive semi-definite, the right side of Eq. (6) is convex and has a unique global minimizer:

$$d_x^* = -\frac{1}{\beta} F^{-1}(\theta_{\text{old}}) \nabla_\theta \mathcal{L}_{\text{policy}}(x; \theta_{\text{old}}). \tag{7}$$

By substituting $d_x^*$ into the Taylor expansion of $\mathcal{L}(x; \theta_{\text{old}} + d)$, we derive the loss descent value:

$$\mathcal{L}(x; \theta_{\text{old}}) - \mathcal{L}(x; \theta_{\text{old}} + d_x^*) = \frac{1}{2\beta} \nabla_\theta \mathcal{L}_{\text{policy}}(x; \theta)\big|_{\theta=\theta_{\text{old}}}^T F^{-1}(\theta_{\text{old}}) \nabla_\theta \mathcal{L}_{\text{policy}}(x; \theta)\big|_{\theta=\theta_{\text{old}}} \tag{8}$$

$$= \frac{1}{2\beta} \nabla_\theta a_\theta(x)\big|_{\theta=\theta_{\text{old}}}^T F^{-1}(\theta_{\text{old}}) \nabla_\theta a_\theta(x)\big|_{\theta=\theta_{\text{old}}}. \tag{9}$$

Finally, note that $r(x, y)$ is an unbiased estimator of $a_\theta(x)$. By applying the vector parameter Cramér–Rao bound to Eq. (9) and summing over $x \sim \mathcal{D}$, we get the upper bound shown in Eq (5) (the equality becomes an inequality because $d$ may not simultaneously satisfy all $d_x^*$). The full derivation is provided in Appendix A. Eq. (5) indicates the convergence rate is correlated with the problem difficulty. The policy model learns little from too easy ($a_\theta(x) \to 1$) or too hard problems ($a_\theta(x) \to 0$), and the maximal efficiency upper bound is achieved at 50% accuracy.

## 4.2 Difficulty-Aware Hint Manipulation via Multi-Round Sampling

From Eq. (5), we have established how problem difficulty affects learning efficiency. A natural follow-up problem is whether we can deliberately adjust the problem difficulty to lie within the high-efficiency region (around 50%). Recent studies (Liu et al., 2025a; Huang et al., 2025b; Yan et al., 2025; Fu et al., 2025) have shown that incorporating off-policy hint guidance into on-policy exploration can enhance the success rate of exploration on challenging samples. However, their

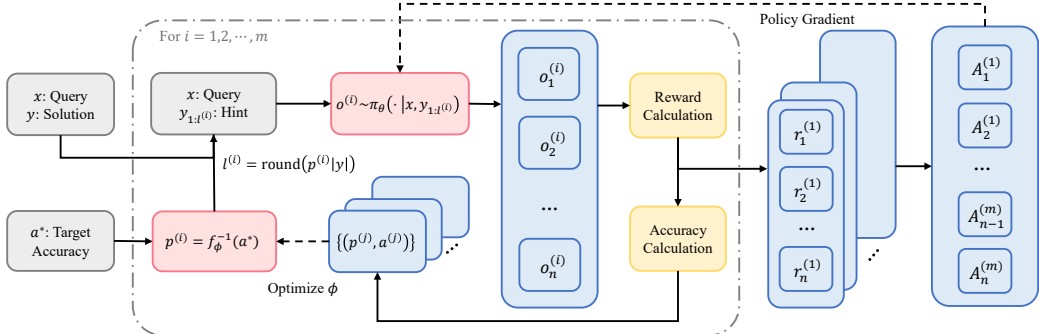

Figure 2: Overview of SEELE. In each step, SEELE conducts $m$ rollout rounds. In round $i$, an adaptive hint $y_{1:l^{(i)}}$ is appended to the original problem, where $y_{1:l^{(i)}}$ is a prefix of the full solution $y$ with length $l^{(i)}$, determined by the accuracy-hint model $f_\phi$ to achieve the target rollout accuracy $a^*$. The output accuracies within each round are then used to update $f_\phi$, enabling more accurate predictions in subsequent rounds. Finally, SEELE computes the advantage function over the outputs of all $m$ rounds, which is used to update the policy model.

strategies for controlling hint length lack a principled objective and fail to consider the instance-level difficulty as well as the model's real-time capability. As a result, the difficulty of hinted problems may deviate from the optimal region, thereby limiting optimization efficiency.

We draw inspiration from these hint-based methods and propose SEELE, a novel instance-level capability-adaptive hint manipulation approach built on GRPO. SEELE adds a dynamic length hint after the original problem to control difficulty to match the increasing model capability, maintaining the prediction accuracy around 50%. To determine the optimal hint length, it is necessary to estimate the difficulty of the problem relative to the model capability at the current timestep. For this purpose, we introduce an accuracy estimator $f_\phi$ that captures the relationship between prediction accuracy and hint length (the specific form of $f_\phi$ and its optimization is introduced in Section 4.3).

As shown in Figure 2, we design a multi-round adaptive sampling framework. Different from standard GRPO rollout generation, SEELE distributes the rollouts into $m$ rounds to gradually fit the parameter $\phi$. In round $i$, SEELE first predicts a hinting rate $p^{(i)}$ for reaching target accuracy $a^*$ by the inverse function of $f_\phi$. Then, the corresponding part of the solution will be concatenated after $x$ as the input of the policy model for generating $n$ outputs $o_1^{(i)}, o_2^{(i)}, \cdots, o_n^{(i)}$. We calculate the accuracy within this round as $a^{(i)}$ and add the current hint-accuracy pair $(p^{(i)}, a^{(i)})$ to the memory and update the parameter of $f_\phi$. As the number of rounds increases, $f_\phi$ will model the accuracy more and more accurately and finally make the rollout accuracy stabilize at $a^*$. Specifically, the first round needs a cold start where we use a default hinting rate $\frac{|y|-1}{|y|}$ in preparation for the worst model capability. The final hinting rate after $m$ rounds will be stored in the dataset so that in the next epoch SEELE can begin exploration from the last predicted rate.

After completing all rollout rounds, the $mn$ outputs will be used to calculate the advantages. In GRPO implementation, the advantage is computed at the response level and then distributed to all tokens, which will undermine the model's output probability on input hints if the model fails to explore a correct completion. Hence, we only compute RLVR loss on the generated tokens and impose a supervised loss on the hint tokens to encourage imitation. The final loss is

$$\mathcal{L}(\theta) = -\mathbb{E}_{x \sim \mathcal{D}, o \sim \pi_\theta(\cdot|\hat{x})}[A_{\theta_{\text{old}}}(\hat{x}, o) + \gamma \log \pi_\theta(y_{1:l}|x)] + \beta \mathbb{E}_{x \sim \mathcal{D}}[\mathbb{D}_{\text{KL}}(\pi_{\theta_{\text{old}}}(\cdot|x)||\pi_\theta(\cdot|x)], \quad (10)$$

where $\hat{x}$ denotes the concatenated input $x \oplus y_{1:l}$ and $y_{1:l}$ is the hint decided by multi-round sampling.

## 4.3 Rollout Accuracy Prediction

The effectiveness of SEELE critically depends on the accuracy of the prediction model $f_\phi$, making its design particularly important. $f_\phi$ should be sufficiently expressive to fit the accuracy-hint relationship while also capable of generalizing from only a few data points. To construct such an accurate yet tractable model, we adopt an established framework used in humans. In psychometrics,

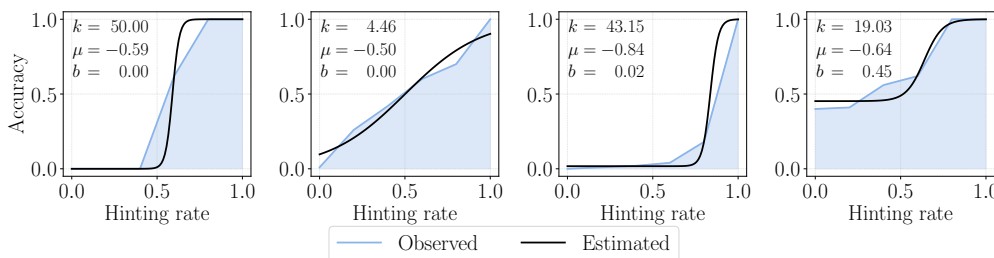

Figure 3: Cases of the accuracy-hint curves and the 3PL fitted curve and parameters. We select 4 typical curves to demonstrate the expressive power of 3PL model.

Item Response Theory (IRT) (Chen et al., 2021) studies the relationship between an individual's performance on a test item and the test takers' levels of performance on an overall measure of the ability. The three-parameter logistic (3PL) model is a widely used IRT model that gives the probability that a person with a given ability level will answer correctly:

$$a(\Theta) = b + \frac{1-b}{1 + e^{-k_0(\Theta - \nu)}}, \tag{11}$$

where $a$ and $\Theta$ indicate the tester's prediction accuracy and capability, and three model parameters $\nu, k_0, b$ represent difficulty, discrimination, and guessing chance, respectively.

In our approach, the difficulty of problem is tunable depending on the hinting rate. We assume the measure of difficulty $\nu$ is linearly related to the hinting rate $p$ as $\nu = -\frac{k}{k_0}p + \nu_0$. Through some symbol substitution and algebraic transformations, we derive the relationship between the accuracy and hinting rate at a certain level of model capability:

$$f_\phi(p) = b + \frac{1-b}{1 + e^{-k(p+\mu)}}, \tag{12}$$

where $\phi$ represents the parameters $\{k, \mu, b\}$ and $\mu = \frac{k_0}{k}(\Theta - \nu_0)$ is the shifted capability measure.

The relationship described in (12) aligns with both our intuition and empirical observations. When $p$ is small, the problem is too difficult for the model to explore a correct reasoning path, resulting in near-zero accuracy. As $p$ increases and critical steps are gradually revealed, the LLM becomes capable of completing the solution, leading to a rapid increase in accuracy. Once all critical steps are revealed, the model's accuracy approaches 1, and providing a longer hint yields no further gain.

Across different problems and training stages, the accuracy-hint curve varies in terms of its starting point, slope, and upper bound. We analyze these curves for 100 randomly sampled problems, with the model generating 100 outputs per problem at each hint level, and find that the 3PL model is sufficiently expressive to capture all observed patterns. Full illustrations are provided in Appendix H, while Figure 3 presents representative examples. The observed trends are consistent with IRT predictions, and the 3PL model accurately captures the relationship. Moreover, as the 3PL model involves only three parameters, it requires only a few rounds to obtain an accurate fit. For the model fitting, SEELE employs non-linear least squares:

$$\phi = \arg\min_\phi \sum_{j=1}^{i} \left( f_\phi(p^{(j)}) - \hat{a}^{(j)} \right)^2, \text{ where } \hat{a}^{(j)} = \text{mean}(\{a^{(w)} | p^{(w)} = p^{(j)}, w = 1, \cdots, i\}). \tag{13}$$

Here $\hat{a}^{(j)}$ denotes the averaged accuracy across all rounds with the same hinting rate, which helps reduce variance in the accuracy estimation.

## 5 EXPERIMENTS

### 5.1 SETUP

**Training Datasets.** We select DeepMath-103K as our training dataset. DeepMath-103K (He et al., 2025) is a large-scale, decontaminated SFT dataset featuring challenging and verifiable mathematical problems, with a strong focus on higher-difficulty problems. To construct a challenging training subset, we filter out 22k problems that are particularly difficult, i.e., those on which

Qwen2.5-7B (Qwen et al., 2025) fails to produce a correct answer. For these problems, we annotate step-by-step reasoning traces using DeepSeek-V3 (DeepSeek-AI et al., 2025b). Detailed data synthesis procedure and the annotation prompt are included in Appendix F.

**Implementation Details.** We adopt GRPO as the RL algorithm, setting the KL coefficient $\beta = 0.001$ and the imitation coefficient $\gamma = 0.001$. Our rollout batch size is 256 and the update batch size is 64. For our approach and all other RL-based baselines, we generate 32 rollouts in total with a maximum length 2,048 tokens for each sample. For our multi-round sampling, we set the number of rounds $m = 4$ and each round will generate $n = 8$ rollouts. Temperature is set to 1.0 for the rollout generation. Our training is based on veRL (Sheng et al., 2025). We use MathRuler (hiyouga, 2025) to verify the correctness of the model's outputs and use the TRF non-linear least squares algorithm provided by the LMFIT library (Newville et al., 2025) to fit $f_\phi$. Our experiments are conducted under the Zero-RL setting, where we use the DeepSeek-R1-Zero prompting template and train the base model for 400 steps. The tested models include Qwen2.5-Math-7B, Qwen2.5-1.5B/3B, LLaMA 3.2-3B, and Mathstral-7B-v0.1. More details are included in Appendix B and G.

**Evaluation.** Following prior works (Zeng et al., 2025; Huang et al., 2025b), we evaluate SEELE on 6 math and 3 general reasoning benchmarks. The math benchmarks are GSM8K (Cobbe et al., 2021), MATH500 (Hendrycks et al., 2021), Minerva (Lewkowycz et al., 2022), OlympiadBench (He et al., 2024), AIME24 (LI et al., 2024), and AMC23 (He et al., 2024). The general benchmarks are ARC-Challenge (Clark et al., 2018), GPQA-Diamond (Rein et al., 2023), and MMLU-Pro (Wang et al., 2024). For AIME24 and AMC23, we report avg@32 due to their small test sets and use a generation temperature of 0.6; for others, we use greedy decoding and report pass@1 accuracy.

**Baselines.** We compare SEELE with pure GRPO and SFT baselines and recent supervision-aided RLVR methods: LUFFY (Yan et al., 2025), UFT (Liu et al., 2025a), and Prefix-RFT (Huang et al., 2025b). Among them, UFT, and Prefix-RFT are concurrent hinted-based approaches. For a fair comparison, we train all the methods on the same dataset, prompt template, and models and set the hyperparameters following their paper.

## 5.2 MAIN RESULTS

Table 1 summarizes the performance on Qwen2.5-Math-7B and Qwen2.5-1.5B, while results on other models are included in Appendix C. SEELE consistently outperforms all baselines, demonstrating clear advantages across tasks and model scales. For the larger Qwen2.5-Math-7B model, SEELE achieves the best math reasoning average of **59.0%**, surpassing the strongest baseline (Prefix-RFT) by nearly 3 points, and also attains the highest general domain average of **58.5%**. As Qwen2.5-Math-7B is a well-trained math model, it benefits a lot from self-exploration and relies relatively less on the off-policy supervision, which accounts for GRPO's large advantage over SFT. While LUFFY, UFT, and Prefix-RFT incorporate the supervision in the on-policy RL, their strategies are too rigid and fail to boost the training, only achieving marginal improvements over GRPO.

Compared with the 7B setting, SEELE shows even more pronounced advantages on the smaller Qwen2.5-1.5B model, where SEELE improves the math average to **34.2%**, achieving an average improvement of **+10.0** points over GRPO, **+8.4** points over SFT, and **+3.8** points over the second best UFT. On the general domain reasoning tasks, SEELE also outperforms the strongest baseline (SFT) by **+2.7** points. The lower intial model capability of Qwen2.5-1.5B underscores the benefits of introducing off-policy demonstrations and leveraging dynamic data difficulty adaptation. We further observe that GRPO generally underperforms SFT on complex reasoning tasks (e.g., MATH500, Minerva, AMC23), while showing comparable or slightly better performance on relatively easy tasks (e.g., GSM8K, ARC-C), demonstrating the limitations of purely on-policy exploration. Moreover, the relatively low performance of GRPO and SFT indicates that exclusive self-exploration or pure imitation alone is insufficient to cultivate strong complex reasoning capabilities.

## 5.3 TRAINING DYNAMICS

Figure 4 shows the training dynamics (reward, response length, and validation accuracy) of the SEELE, GRPO, and the two dynamic hinting baselines using Qwen2.5-3B.

**Precise Difficulty Control** After the initial warm-up, the reward of SEELE rapidly rises to around 0.5 and maintains throughout the remaining training process, indicating our multi-round sampling and regression framework is able to precisely control the rollout accuracy. Due to the adoption of

Table 1: Accuracy on math and general domain reasoning benchmarks.

| Model | Math Reasoning | | | | | | | General Domain Reasoning | | | |
|---|---|---|---|---|---|---|---|---|---|---|---|
| | GSM8K | MATH500 | Minerva | Olympiad | AIME24 | AMC23 | Avg. | ARC-C | GPQA-D | MMLU-Pro | Avg. |
| **Qwen2.5-Math-7B** | 71.6 | 63.2 | 25.7 | 32.0 | 14.1 | 45.2 | 42.0 | 69.5 | 24.7 | 17.7 | 37.3 |
| + SFT | 89.5 | 74.6 | 35.7 | 37.9 | 12.4 | 52.2 | 50.4 | 77.6 | 37.4 | 47.9 | 54.3 |
| + GRPO | 92.0 | 80.6 | 36.0 | 41.2 | 23.9 | 60.2 | 55.7 | 77.1 | 37.4 | 45.2 | 53.2 |
| + LUFFY | 91.7 | 80.0 | 35.3 | 42.4 | 18.5 | 66.2 | 55.7 | 80.5 | 39.9 | 49.9 | 56.8 |
| + UFT | 92.1 | 82.4 | 34.6 | 40.3 | 17.6 | 66.6 | 55.6 | **81.0** | 40.9 | 49.9 | 57.3 |
| + Prefix-RFT | 92.1 | 81.6 | 36.8 | 43.0 | 20.9 | 63.5 | 56.3 | 80.3 | 37.4 | 49.8 | 55.8 |
| + SEELE (Ours) | **92.4** | **82.6** | **37.1** | **46.5** | **25.8** | **69.7** | **59.0** | 80.7 | **42.9** | **52.0** | **58.5** |
| **Qwen2.5-1.5B** | 61.9 | 22.8 | 9.6 | 6.7 | 0.7 | 9.1 | 18.5 | 45.1 | 15.7 | 12.2 | 24.3 |
| + SFT | 67.4 | 43.6 | 13.6 | 12.6 | 1.4 | 16.4 | 25.8 | 63.7 | 25.8 | 30.2 | 39.9 |
| + GRPO | 70.1 | 36.4 | 10.7 | 11.1 | 1.8 | 15.2 | 24.2 | 64.8 | 25.3 | 23.1 | 37.7 |
| + LUFFY | 67.2 | 45.4 | 11.0 | 12.9 | 1.6 | 16.5 | 25.8 | 64.3 | 26.8 | 24.7 | 38.6 |
| + UFT | 72.6 | 50.4 | 12.9 | 15.9 | 3.9 | 26.6 | 30.4 | 66.1 | 23.7 | 28.5 | 39.4 |
| + Prefix-RFT | 71.5 | 48.0 | 13.6 | 14.5 | 2.1 | 22.7 | 28.7 | 65.3 | 23.7 | 25.0 | 38.0 |
| + SEELE (Ours) | **76.5** | **58.0** | **16.2** | **19.9** | **4.1** | **30.4** | **34.2** | **68.3** | **27.8** | **31.7** | **42.6** |

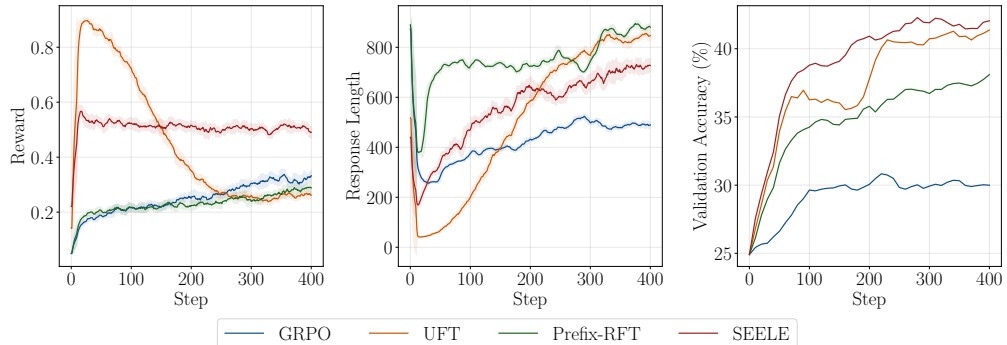

Figure 4: Training dynamics of RL compared with baselines on Qwen2.5-3B. **Left**: training rewards; **Middle**: Response length; **Right**: averaged accuracy on math reasoning validation sets.

a pessimistic cold-start strategy, the initial reward exceeds the target 0.5 and gradually decreases as training progresses. To further understand SEELE's difficulty control mechanism, we visualize the intermediate results from the multi-round rollout sampling process in Figure 5. Relatively larger fluctuations and deviations are observed in Rounds 1 and 2, primarily because the 3PL regression model receives too few samples to produce accurate fits. Particularly, there is a peak in the first 80 steps (corresponding to the first epoch), which is caused by the high cold-start hinting rate. From the second epoch onward, SEELE uses the hinting rate rectified in the preceding epoch for the initial round, resulting in more accurate predictions. By Round 3 and Round 4, the accuracy is very close to the target 50%, suggesting that the minimal data requirement for the 3PL model is approximately three samples, while four samples are sufficient to fit a model with adequate precision in practice.

**Accelerated Learning** Compared with GRPO, UFT, and Prefix-RFT, SEELE consistently achieves higher accuracy throughout the entire training process and converges more rapidly. Within the first 100 steps, both SEELE and UFT quickly widen the accuracy gap over GRPO and Prefix-RFT, underscoring the importance of external guidance. GRPO reaches its performance ceiling around step 100, showing minimal improvement thereafter. Notably, its reward continues to increase throughout the remainder of training and ultimately surpasses that of UFT and Prefix-RFT. This behavior suggests that GRPO primarily reinforces previously acquired skills rather than facilitating the learning of new capabilities. From the 100 to 200 steps, the performance of UFT stagnates because its problem difficulty fails to adapt to the model's evolving capabilities. UFT's holistic pre-designed decaying schedule cannot adequately meet the changing learning requirements. In contrast, SEELE sustains a high growth rate until approximately step 300. From the perspective of response length, SEELE exhibits a more stable growth trend than UFT and Prefix-RFT, which aligns with the performance growth, highlighting the effectiveness of our hint adjustment schedule.

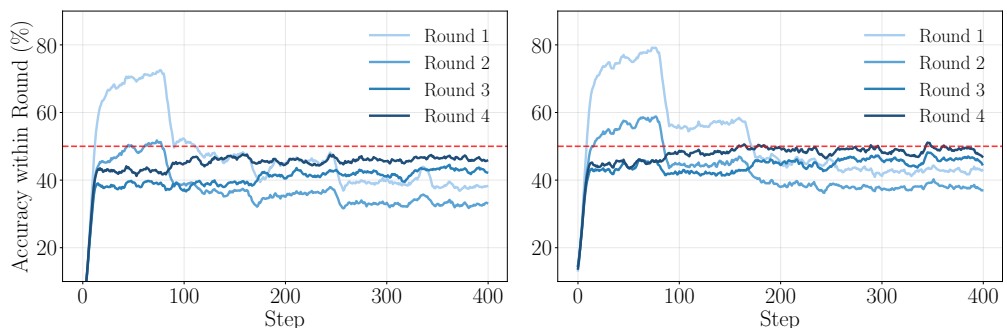

Figure 5: Accuracy within each rollout round during training. The red dotted line denotes the targeted accuracy. **Left**: Qwen2.5-1.5B; **Right**: Qwen2.5-3B.

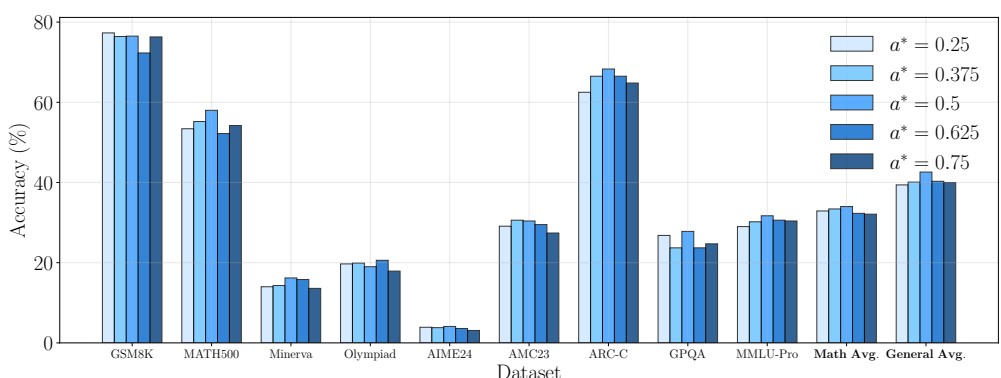

Figure 6: Performance for different target accuracy $a^*$ using Qwen2.5-1.5B.

## 5.4 TARGET ACCURACY ANALYSIS

To verify the critical role of rollout accuracy in RLVR, we conduct an ablation study by varying the target accuracy $\alpha^*$ and examining its impact on performance. Specifically, we set $\alpha^*$ to values in the range $\{i/n \mid i = 2, 3, \ldots, n-2\}$, while excluding the boundary cases $i = 1$ and $i = n-1$ to avoid situations where rollouts become entirely incorrect or entirely correct. The results, summarized in Figure 6, lead to the following observations: (1) Setting the target accuracy to $0.5$ yields the best performance, while performance degrades gradually as $\alpha^*$ deviates from this value; (2) The degradation trend is approximately symmetric, whether the target accuracy is increased or decreased. These findings are consistent with both intuition and theoretical analysis: data that is either overly difficult or overly simple hinders effective training. For completeness, we also report detailed accuracy control results in Appendix D, which further demonstrate that our method can precisely regulate the rollout accuracy to any desired target value.

## 6 CONCLUSION

In this paper, we present SEELE, a novel reinforcement learning with variable reward (RLVR) framework that leverages off-policy demonstrations to dynamically align problem difficulty with the evolving capability of the model, thereby optimizing training efficiency. This framework is motivated by our quantitative analysis, which shows that the learning efficiency of RL algorithms is maximized when the policy model's rollout accuracy is approximately 50%. A key distinguishing feature of our approach, compared with previous supervision-aided RL methods, is that the difficulty manipulation operates at the instance level and incorporates real-time feedback, rendering the training process more responsive and enhancing the utility of each individual training sample. Extensive experiments demonstrate that SEELE significantly outperforms GRPO, SFT, and other RLVR baselines. Our study provides preliminary insights into the types of data favored by RL algorithms and offers a novel direction for improving data efficiency in reinforcement learning.

## REPRODUCIBILITY STATEMENT

The SEELE algorithm is described in Section 4.2 and 4.3. We also provide a detailed pseudocode in Appendix B. The experimental setup is mainly described in Section 5.1 and the prompts used for data annotation and RL training are provided in Appendix F and G. For each baseline, we set the unspecified hyperparameters following the reported values in their paper. To facilitate reproduction, we release the code and data at https://anonymous.4open.science/r/seele-81BC.

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

## LLM Usage

We only use large language models for grammar checking and expression polishing, without involving any research uses such as finding related work, research ideation, and experimental design. All research concepts, ideas, and analyses were developed and conducted by the authors. We provide the original manuscript to OpenAI GPT-4o and instruct it to *check the errors and make the given text more professional, coherent, and native as part of a research paper*.

We have ensured that the LLM-generated text adheres to ethical guidelines and does not contribute to plagiarism or scientific misconduct. All generated text is carefully reviewed to confirm factual accuracy and fidelity to the original content.

## A  Derivation of the RL Optimization Efficiency and Rollout Accuracy

We start from Eq. (1) and assume the reference distribution $\pi_{\text{ref}}$ equals $\pi_{\text{old}}$. Considering a one-step gradient update with the update vector $d$, we substitute $\theta = \theta_{\text{old}} + d$ into the loss function and obtain the new loss value as

$$\mathcal{L}(\theta_{\text{old}} + d) = \mathcal{L}_{\text{policy}}(\theta_{\text{old}} + d) + \beta \mathbb{E}_{x \sim \mathcal{D}} \left[ \mathbb{D}_{\text{KL}} \left( \pi_{\theta_{\text{old}}}(\cdot|x) || \pi_{\theta_{\text{old}}+d}(\cdot|x) \right) \right]. \tag{14}$$

Note that $\mathcal{L}$ is defined as the independent sum of loss on each prompt $x$. So, we can compute the individual loss for each $x$ and then aggregate the results. For a specific $x$, we approximate $\mathcal{L}_{\text{policy}}(x; \theta_{\text{old}} + d)$ and $\mathbb{D}_{\text{KL}} \left( \pi_{\theta_{\text{old}}}(\cdot|x) || \pi_{\theta_{\text{old}}+d}(\cdot|x) \right)$ using first-order and second-order Taylor expansion at $\theta_{\text{old}}$, respectively. First, we expand the policy loss:

$$\mathcal{L}_{\text{policy}}(x; \theta_{\text{old}} + d) \approx \mathcal{L}_{\text{policy}}(x; \theta_{\text{old}}) + \nabla_\theta \mathcal{L}_{\text{policy}}(x; \theta) \big|_{\theta=\theta_{\text{old}}}^T d. \tag{15}$$

$$\begin{aligned}
\mathbb{D}_{\text{KL}} \left( \pi_{\theta_{\text{old}}}(\cdot|x) || \pi_{\theta_{\text{old}}+d}(\cdot|x) \right) = &\mathbb{D}_{\text{KL}} \left( \pi_{\theta_{\text{old}}}(\cdot|x) || \pi_{\theta_{\text{old}}}(\cdot|x) \right) \\
&+ \nabla_\theta \mathbb{D}_{\text{KL}} \left( \pi_{\theta_{\text{old}}}(\cdot|x) || \pi_\theta(\cdot|x) \right) \big|_{\theta=\theta_{\text{old}}}^T d \\
&+ \frac{1}{2} d^T \nabla_\theta^2 \mathbb{D}_{\text{KL}} \left( \pi_{\theta_{\text{old}}}(\cdot|x) || \pi_\theta(\cdot|x) \right) \big|_{\theta=\theta_{\text{old}}} d.
\end{aligned} \tag{16}$$

Next, we compute the first and second order derivatives of KL-divergence.

$$\nabla_\theta \mathbb{D}_{\text{KL}} \left( \pi_{\theta_{\text{old}}}(\cdot|x) || \pi_\theta(y|x) \right) = \nabla_\theta \mathbb{E}_{y \sim \pi_{\theta_{\text{old}}}(\cdot|x)} \log \frac{\pi_{\theta_{\text{old}}}(y|x)}{\pi_\theta(y|x)} \tag{17}$$

$$= -\nabla_\theta \mathbb{E}_{y \sim \pi_{\theta_{\text{old}}}(\cdot|x)} \log \pi_\theta(y|x) \tag{18}$$

$$= -E_{y \sim \pi_{\theta_{\text{old}}}(\cdot|x)} \frac{\nabla_\theta \pi_\theta(y|x)}{\pi_\theta(y|x)} \tag{19}$$

$$= -\sum_y \frac{\pi_{\theta_{\text{old}}}(y|x)}{\pi_\theta(y|x)} \nabla_\theta \pi_\theta(y|x). \tag{20}$$

$$\nabla_\theta \mathbb{D}_{\text{KL}} \left( \pi_{\theta_{\text{old}}}(\cdot|x) || \pi_\theta(y|x) \right) \big|_{\theta=\theta_{\text{old}}} = \sum_y \frac{\pi_{\theta_{\text{old}}}(y|x)}{\pi_{\theta_{\text{old}}}(y|x)} \nabla_\theta \pi_\theta(y|x) \big|_{\theta=\theta_{\text{old}}} \tag{21}$$

$$= \nabla_\theta \left[ \sum_y \pi_\theta(y|x) \right] \Bigg|_{\theta=\theta_{\text{old}}} \tag{22}$$

$$= 0. \tag{23}$$

$$\nabla_\theta^2 \mathbb{D}_{\text{KL}} \left( \pi_{\theta_{\text{old}}}(\cdot|x) || \pi_\theta(\cdot|x) \right) = \nabla_\theta^2 \mathbb{E}_{y \sim \pi_{\theta_{\text{old}}}(\cdot|x)} \log \frac{\pi_{\theta_{\text{old}}}(y|x)}{\pi_\theta(y|x)} \tag{24}$$

$$= -\nabla_\theta^2 \mathbb{E}_{y \sim \pi_{\theta_{\text{old}}}(\cdot|x)} \log \pi_\theta(y|x) \tag{25}$$

$$= -E_{y \sim \pi_{\theta_{\text{old}}}(\cdot|x)} \left[ \frac{\nabla_\theta^2 \pi_\theta(y|x)}{\pi_\theta(y|x)} - \frac{\nabla_\theta \pi_\theta(y|x) \nabla_\theta \pi_\theta(y|x)^T}{\pi_\theta^2(y|x)} \right] \tag{26}$$

$$= E_{y \sim \pi_{\theta_{\text{old}}}(\cdot|x)} \left[ -\frac{\nabla_\theta^2 \pi_\theta(y|x)}{\pi_\theta(y|x)} + \frac{\nabla_\theta \pi_\theta(y|x)}{\pi_\theta(y|x)} \frac{\nabla_\theta \pi_\theta(y|x)^T}{\pi_\theta(y|x)} \right] \tag{27}$$

$$= E_{y \sim \pi_{\theta_{\text{old}}}(\cdot|x)} \left[ -\frac{\nabla_\theta^2 \pi_\theta(y|x)}{\pi_\theta(y|x)} + \nabla_\theta \log \pi_\theta(y|x) \nabla_\theta \log \pi_\theta(y|x)^T \right] \tag{28}$$

$$= E_{y \sim \pi_{\theta_{\text{old}}}(\cdot|x)} \left[ -\frac{\nabla_\theta^2 \pi_\theta(y|x)}{\pi_\theta(y|x)} \right] + F(\theta). \tag{29}$$

$$\tag{30}$$

We substitute $\theta = \theta_{\text{old}}$ for the first term.

$$E_{y \sim \pi_{\theta_{\text{old}}}(\cdot|x)} \left[ \frac{\nabla_\theta^2 \pi_\theta(y|x)}{\pi_\theta(y|x)} \right] \bigg|_{\theta=\theta_{\text{old}}} = \sum_y \frac{\pi_{\theta_{\text{old}}}(y|x)}{\pi_{\theta_{\text{old}}}(y|x)} \nabla_\theta^2 \pi_\theta(y|x) \big|_{\theta=\theta_{\text{old}}} \tag{31}$$

$$= \nabla_\theta^2 \left[ \sum_y \pi_\theta(y|x) \right] \bigg|_{\theta=\theta_{\text{old}}} \tag{32}$$

$$= 0. \tag{33}$$

So, we get

$$\mathcal{L}(x; \theta_{\text{old}} + d) \approx \mathcal{L}_{\text{policy}}(x; \theta_{\text{old}}) + \nabla_\theta \mathcal{L}_{\text{policy}}(x; \theta) \big|_{\theta=\theta_{\text{old}}}^T d + \frac{\beta}{2} d^T F(\theta_{\text{old}}) d. \tag{34}$$

Since the Fisher information matrix is positive semi-definite, the right hand of Eq. (34) convex has a unique global minimizer. To find the minimizer $d_x^*$, let the derivative be zero.

$$\nabla_\theta \mathcal{L}_{\text{policy}}(x; \theta) \big|_{\theta=\theta_{\text{old}}} + \beta F(\theta_{\text{old}}) d_x^* = 0. \tag{35}$$

$$d_x^* = -\frac{1}{\beta} F(\theta_{\text{old}})^{-1} \nabla_\theta \mathcal{L}_{\text{policy}}(x; \theta) \big|_{\theta=\theta_{\text{old}}}. \tag{36}$$

Substitute back to Eq. (34)

$$\mathcal{L}(x; \theta_{\text{old}}) - \mathcal{L}(x; \theta_{\text{old}} + d_x^*) \approx \frac{1}{2\beta} \nabla_\theta \mathcal{L}_{\text{policy}}(x; \theta) \big|_{\theta=\theta_{\text{old}}}^T F^{-1}(\theta_{\text{old}}) \nabla_\theta \mathcal{L}_{\text{policy}}(x; \theta) \big|_{\theta=\theta_{\text{old}}}. \tag{37}$$

Now, we need to connect $\nabla_\theta \mathcal{L}_{\text{policy}}(x; \theta) \big|_{\theta=\theta_{\text{old}}}$ with $a_{\theta_{\text{old}}}(x)$. According to the definition

$$\nabla_\theta \mathcal{L}_{\text{policy}}(x; \theta) = \nabla_\theta \mathbb{E}_{y \sim \pi_\theta(\cdot|x)} A_{\theta_{\text{old}}}(x, y) \tag{38}$$

$$= \nabla_\theta \mathbb{E}_{y \sim \pi_\theta(\cdot|x)} \left[ r(x, y) - \mathbb{E}_{y \sim \pi_{\theta_{\text{old}}}(\cdot|x)} r(x, y) \right] \tag{39}$$

$$= \nabla_\theta \mathbb{E}_{y \sim \pi_\theta(\cdot|x)} [r(x, y)] \tag{40}$$

$$= \nabla_\theta a_{\theta_{\text{old}}}(x). \tag{41}$$

$$\mathcal{L}(x; \theta_{\text{old}}) - \mathcal{L}(x; \theta_{\text{old}} + d_x^*) \approx \frac{1}{2\beta} \nabla_\theta a_\theta(x) \big|_{\theta=\theta_{\text{old}}}^T F^{-1}(\theta_{\text{old}}) \nabla_\theta a_\theta(x) \big|_{\theta=\theta_{\text{old}}}. \tag{42}$$

Note that $r(x, y)$ is an unbiased estimator of $a_\theta(x)$. By applying the vector parameter Cramér–Rao bound (Kay, 1993, Section 3.8) to Eq. (42), we get

$$\mathcal{L}(x; \theta_{\text{old}}) - \mathcal{L}(x; \theta_{\text{old}} + d_x^*) \approx \frac{1}{2\beta} \text{Var}_{y \sim \pi_{\theta_{\text{old}}}(\cdot|x)} [r(x, y)] \tag{43}$$

$$= \frac{1}{2\beta} a_{\theta_{\text{old}}}(x)(1 - a_{\theta_{\text{old}}}(x)). \tag{44}$$

---

**Algorithm 1** SEELE Policy Optimization

---

**Input:** initial policy model $\pi_{\text{init}}$; training set $\mathcal{D}$

1: policy model $\pi_\theta \leftarrow \pi_{\text{init}}$
2: reference model $\pi_{\text{ref}} \leftarrow \pi_{\text{init}}$
3: **for** step=$1, 2, \cdots, T$ **do**
4:      $\pi_{\theta_{\text{old}}} \leftarrow \pi_\theta$
5:      Sample a batch $B$ from $\mathcal{D}$
6:      $C \leftarrow \{x : \emptyset\}_{x \in B}$    (Here $C_x$ refers to $\{(p_x^{(j)}, a_x^{(j)})\}_{j=1}^{i-1}$)
7:      **for** $i = 1, 2, \cdots, m$ **do**
8:          $B_i \leftarrow \emptyset$
9:          **for** $x, y \in B$ **do**
10:              $\hat{C}_x \leftarrow C_x \cup \{(w, w) | w \in \{0, 1\} \setminus \{p_x^{(j)}\}_{j=1}^{i-1}\}$
11:              Optimize $\phi$ on $\hat{C}_x$ (Follow Eq.(13).)
12:              $p_x^{(i)} = f_\phi^{-1}(a^*)$
13:              $\hat{x} = x \oplus y_{1:l}, \quad l = \text{round}(p_x^{(i)}|y|)$
14:              $B_i \leftarrow B_i \cup \{(\hat{x}, y)\}$
15:          **end for**
16:          Sample $n$ outputs $\{o_j^{(i)}\}_{j=1}^n \sim \pi_{\theta_{\text{old}}}(\cdot|\hat{x})$ for each $\hat{x} \in B_i$
17:          Compute rewards $\{r_j^{(i)}\}_{j=1}^n$ for each sampled output
18:          Compute rollout accuracy $a_x^{(i)}$ over $\{r_j^{(i)}\}_{j=1}^n$ for each problem $x$
19:          **for** $x \in B$ **do**
20:              $C_x \leftarrow C_x \cup \{(p_x^{(i)}, a_x^{(i)})\}$
21:          **end for**
22:      **end for**
23:      Compute advantage $\{A_{j,t}^{(i)}\}_{i=j=1}^{m,n}$ for each sampled token
24:      **for** GRPO iteration=$1, 2 \cdots, u$ **do**
25:          Update the policy model $\pi_\theta$ by maximizing the objective in Eq.(47).
26:      **end for**
27: **end for**
**Output:** $\pi_\theta$

---

So, for an arbitrary $d$

$$\mathcal{L}(x; \theta_{\text{old}}) - \mathcal{L}(x; \theta_{\text{old}} + d) \leq \frac{1}{2\beta} a_{\theta_{\text{old}}}(x)(1 - a_{\theta_{\text{old}}}(x)). \tag{45}$$

Now we have got the equality for a single $x$, taking the expectation we get

$$\mathcal{L}(\theta_{\text{old}}) - \mathcal{L}(\theta_{\text{old}} + d) \leq \frac{1}{2\beta} \mathbb{E}_{x \sim \mathcal{D}}[a_{\theta_{\text{old}}}(x)(1 - a_{\theta_{\text{old}}}(x))]. \tag{46}$$

## B   POLICY OPTIMIZATION IMPLEMENTATION DETAILS

In the practical implementation, we apply the clipping techniques used in GRPO (Shao et al., 2024) and use $\pi_{ref}$ rather than $\pi_{\text{old}}$ in KL regularization for simplicity, as shown in Eq. (48) and Eq. (47), which slightly differs from the formulation in Eq. (10).

$$\mathcal{L}(\theta) = -\mathbb{E}_{x \sim \mathcal{D}, o \sim \pi_\theta(\cdot|\hat{x})}[\hat{A}_{\theta_{\text{old}}}(\hat{x}, o) + \gamma \log \pi_\theta(y_{1:l}|x)] + \beta \mathbb{E}_{x \sim \mathcal{D}}[\mathbb{D}_{\text{KL}}(\pi_{\theta_{\text{ref}}}(\cdot|x)||\pi_\theta(\cdot|x)), \tag{47}$$

$$\hat{A}_{\theta_{\text{old}}}(\hat{x}, o) = \sum_{t=1}^o \left\{ \min\left[ \frac{\pi_\theta(o_t|\hat{x}, o_{<t})}{\pi_{\theta_{\text{old}}}(o_t|\hat{x}, o_{<t})} A_t, \text{clip}\left( \frac{\pi_\theta(o_t|\hat{x}, o_{<t})}{\pi_{\theta_{\text{old}}}(o_t|\hat{x}, o_{<t})}, 1 - \epsilon, 1 + \epsilon \right) A_t \right] \right\} \tag{48}$$

Table 2: Performance on math and general domain reasoning benchmarks using Qwen2.5-3B, LLaMA 3.2-3B, and Mathstral-7B-v0.1.

| Model | Math Reasoning | | | | | | | General Domain Reasoning | | | |
| --- | --- | --- | --- | --- | --- | --- | --- | --- | --- | --- | --- |
| | GSM8K | MATH500 | Minerva | Olympiad | AIME24 | AMC23 | Avg. | ARC-C | GPQA-D | MMLU-Pro | Avg. |
| **Qwen2.5-3B** | 73.3 | 29.0 | 7.0 | 10.7 | 0.6 | 11.6 | 22.0 | 66.6 | 23.7 | 15.2 | 35.2 |
| + SFT | 73.1 | 52.6 | 18.4 | 18.5 | 2.3 | 25.0 | 31.7 | 75.7 | 30.3 | 40.1 | 48.7 |
| + GRPO | 78.5 | 45.8 | 17.3 | 15.0 | 1.7 | 23.9 | 30.4 | 75.6 | 36.9 | 40.5 | 51.0 |
| + LUFFY | 81.1 | 56.6 | 21.0 | 21.3 | 2.7 | 27.0 | 35.0 | 78.4 | 32.8 | 41.5 | 50.9 |
| + UFT | 84.8 | 62.6 | 22.1 | 25.9 | 5.8 | **41.6** | 40.5 | 79.4 | 35.4 | 42.9 | 52.6 |
| + Prefix-RFT | 77.4 | 57.0 | 21.3 | 21.9 | **6.0** | 31.5 | 35.9 | **82.2** | **35.9** | 37.9 | 52.0 |
| + SEELE (Ours) | **86.3** | **66.4** | **26.1** | **28.9** | 5.9 | 39.4 | **42.2** | 81.2 | 34.3 | **44.0** | **53.2** |
| **LLaMA 3.2-3B** | 8.1 | 4.2 | 2.6 | 1.9 | 0.0 | 2.7 | 3.3 | 13.7 | 15.2 | 7.3 | 12.1 |
| + SFT | 20.7 | 8.8 | 4.0 | 3.0 | 0.2 | 4.0 | 6.8 | 59.0 | 27.3 | 26.3 | 37.5 |
| + GRPO | 5.9 | 8.2 | 5.5 | 2.5 | 0.0 | 3.2 | 4.2 | 43.5 | 28.8 | 23.9 | 32.1 |
| + LUFFY | 11.9 | 9.6 | 4.8 | 2.4 | 0.2 | 3.5 | 5.4 | 58.8 | 30.8 | 26.0 | 38.5 |
| + UFT | 24.2 | 11.4 | 8.1 | 3.6 | 0.2 | **7.5** | 9.2 | 56.2 | 27.3 | 25.2 | 36.2 |
| + Prefix-RFT | 21.1 | 9.2 | 5.8 | 3.6 | 0.1 | 2.8 | 7.1 | 55.5 | 28.4 | 24.9 | 36.3 |
| + SEELE (Ours) | **29.5** | **12.8** | **7.4** | **5.2** | **0.7** | 6.5 | **10.4** | 62.4 | 28.8 | 26.1 | **39.1** |
| **Mathstral-7B** | 76.0 | 34.6 | 15.8 | 14.5 | 1.4 | 15.7 | 26.3 | 62.9 | 29.3 | 16.7 | 36.3 |
| + SFT | 80.1 | 53.4 | 24.6 | 21.6 | 1.8 | 26.6 | 34.7 | 73.1 | **44.4** | 42.2 | 53.2 |
| + GRPO | 85.6 | 44.0 | 22.4 | 16.0 | 0.9 | 21.7 | 31.8 | 80.0 | 38.4 | 43.5 | 54.0 |
| + LUFFY | 88.4 | 60.0 | 25.0 | 24.4 | 6.4 | 32.4 | 39.4 | 80.2 | 32.8 | 44.1 | 52.4 |
| + UFT | 87.9 | 57.6 | 23.2 | 20.6 | 4.3 | 33.1 | 37.8 | 81.8 | 37.9 | 47.7 | 55.8 |
| + Prefix-RFT | 86.3 | 59.6 | 24.3 | 22.5 | 3.6 | 31.7 | 38.0 | 79.7 | 35.4 | 46.2 | 53.8 |
| + SEELE (Ours) | **90.2** | **63.4** | **27.9** | **29.5** | **6.7** | **38.4** | **42.7** | **82.9** | 38.9 | **50.5** | **57.4** |

In Algorithm 1, we present the detailed workflow of SEELE policy optimization. At each training step, SEELE first samples a batch from the training set and constructs a hint–accuracy collection $C_x$ for each problem $x$. Then, SEELE performs $m$ rounds of generation. In each round, $C_x$ is augmented by adding the margin points $(0, 0)$ and $(1, 1)$ whenever $p_x = 0$ or $p_x = 1$ has not yet been evaluated. This augmentation substantially enhances fitting accuracy and stability during the early rounds. Subsequently, the predictor $f_\phi$ is optimized on $C_x$ and used to estimate the optimal hint length $l$. With the estimated hint length, SEELE invokes the policy model to generate $n$ rollouts for the hinted problem $\hat{x} = x \oplus y_{1:l}$ and computes the corresponding rewards. These rewards are then aggregated to calculate the intra-round accuracy $a_x$, which is used to update $C_x$. Finally, after $m$ rounds of rollouts, SEELE computes the advantages over all $mn$ outputs following the Dr.GRPO formulation (Liu et al., 2025b), and updates the policy model according to Eq. (47).

## C PERFORMANCE ON VARIOUS MODELS

In addition to Qwen2.5-Math-7B and Qwen2.5-1.5B, we further evaluate SEELE on three base models from different families: Qwen2.5-3B, LLaMA 3.2-3B, and Mathstral-7B-v0.1, and summarize the results in Table 2. Overall, SEELE consistently delivers the strongest performance across both the math reasoning and general domain reasoning benchmarks, demonstrating its wide applicability.

**Qwen2.5-3B.** In the in-domain mathematical reasoning setting, SEELE achieves an average improvement of **+11.8** points over GRPO and **+10.5** points over SFT, surpassing the best baseline by 1.7%. The improvement is consistent across all six math benchmarks, underscoring the benefits of introducing off-policy demonstrations and dynamically adapting data difficulty. We further observe that GRPO generally underperforms SFT on complex reasoning tasks (e.g., MATH500, AIME24), while showing comparable or slightly better performance on relatively easy tasks (e.g., GSM8K, ARC-C), highlighting the limitations of purely on-policy exploration. Moreover, the relatively low performance of GRPO and SFT indicates that exclusive self-exploration or pure imitation alone is insufficient to cultivate strong complex reasoning capabilities. In the out-of-domain general reasoning setting, SEELE demonstrates strong generalization, achieving an average improvement of **+2.2** points over GRPO, while other baselines exhibit only comparable performance.

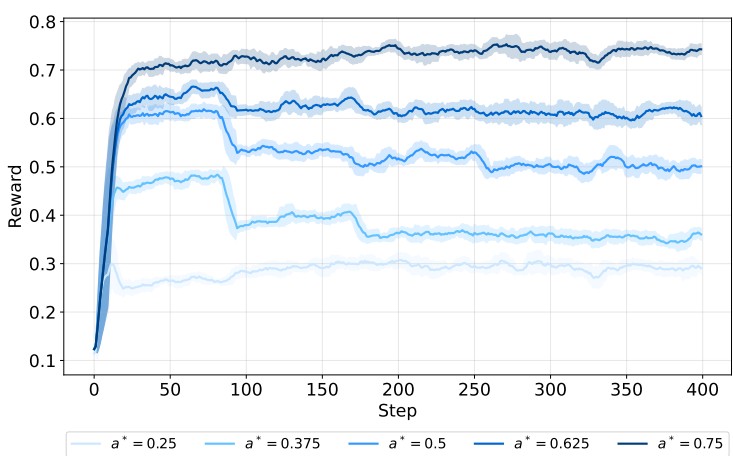

Figure 7: Reward across the training steps for various target accuracy.

**LLaMA 3.2-3B.** A similar trend is observed on the smaller LLaMA 3.2-3B model, where overall math reasoning is more challenging. Despite the low base performance (3.3 average), SEELE improves the math average to **10.4**, outperforming SFT and GRPO by **+3.3** and **+6.2** points respectively. Notably, GRPO again fails to consistently surpass SFT, confirming that naive on-policy reinforcement struggles with scarce reasoning signals. Other supervision-aided RL methods (LUFFY, UFT, Prefix-RFT) provide moderate gains, showing incompetence on the weak initial model. For general domain reasoning, SEELE also reaches the highest average (39.1), exceeding GRPO by **+7.0** and SFT by **+1.6**, showing that our strategy scales effectively even for smaller and weaker base models.

**Mathstral-7B.** On the larger Mathstral-7B, which already exhibits strong math ability, SEELE continues to be the state-of-the-art. It raises the math average to **42.7%**, surpassing SFT by **+8.0** and GRPO by **+10.9** points, and outperforming the strongest competing RL-with-supervision baseline (LUFFY) by **+3.3** points. In challenging tasks such as Olympiad and AIME24, SEELE achieves the largest absolute gains, demonstrating its ability to handle complex reasoning even when the base model is powerful. For general domain reasoning, SEELE further improves the average to **57.4%**, a **+1.6%** gain over the best baseline, again highlighting superior out-of-domain generalization.

In summary, SEELE is generally effective for training models of different families (Qwen, LLaMA, and Mathstral) and sizes (1.5B, 3B, and 7B), showing great robustness. Notably, for models with very weak math ability (e.g., LLaMA 3.2-3B), SEELE still achieves considerable improvement.

## D ROLLOUT ACCURACY MANIPULATION

To verify that our multi-round sampling framework can arbitrarily manipulate the rollout accuracy, we set the target accuracy to $i/n, i = 2, 3, \cdots, n - 2$. We exclude $1/n$ and $(n - 1)/n$ because they are too close to the boundary, which easily leads to all wrong/correct rollout sampling. We set $n = 8$ following the main experiment setup. The results are shown in Table 7. After the cold start for an epoch, the reward converges to the target accuracy across all settings with very little error (less than 0.02 for $a^* > 0.25$)), showing a stable trend until the end of the training. The fluctuation and deviation diminish as the training progresses. The results of $a^* = 0.25$ are slightly higher than the target because they touch the difficulty lower bound, where the policy model can generate sufficiently correct outputs without the aid of hints.

## E ROLLOUT SCHEME ANALYSIS

Given a fixed total number of rollouts, the number of rounds can be configured in multiple ways. Increasing the number of rounds provides more samples for regression, but it reduces the number of samples per round, which in turn increases the variance in estimating accuracy for a specific hinting rate. To examine the actual effect of the multi-round rollout configuration and identify the optimal

Table 3: Performance of different multi-round configurations using Qwen2.5-1.5B.

| Multi-Round Rollout Scheme | | Math Reasoning | | | | | | | General Domain Reasoning | | | |
|---|---|---|---|---|---|---|---|---|---|---|---|---|
| | | GSM8K | MATH500 | Minerva | Olympiad | AIME24 | AMC23 | Avg. | ARC-C | GPQA-D | MMLU-Pro | Avg. |
| $mn = 16$ | $m = 4$ | 76.8 | 51.0 | 15.4 | 19.0 | 3.3 | 28.4 | 32.3 | 66.3 | 24.7 | 29.7 | 40.2 |
| $mn = 24$ | $m = 3$ | 75.1 | 53.4 | 14.7 | **20.0** | **4.6** | 31.8 | 33.3 | 67.9 | 21.9 | 29.8 | 39.9 |
| | $m = 4$ | 77.1 | 54.4 | **16.9** | 19.9 | 3.0 | **32.1** | 33.9 | 66.6 | 24.7 | **33.2** | 41.5 |
| | $m = 6$ | **77.6** | 53.8 | 15.8 | 18.5 | 2.9 | 30.5 | 33.2 | 68.9 | 23.8 | 33.0 | 41.9 |
| $mn = 32$ | $m = 4$ | 76.5 | **58.0** | 16.2 | 19.9 | 4.1 | 30.4 | **34.2** | 68.3 | **27.8** | 31.7 | **42.6** |
| | $m = 8$ | 76.9 | 54.2 | 16.5 | 19.3 | 3.3 | 31.4 | 33.6 | **69.8** | 23.2 | 32.4 | 41.8 |

setting, we train Qwen2.5-1.5B with varying numbers of rollout rounds under different total rollout budgets. The results are presented in Table 3. Our observations are as follows: (1) The performance of SEELE is relatively insensitive to the specific rollout configuration; (2) A three-round scheme yields the lowest performance given the same total number of rollouts, as only two sample points are available for constructing the three-parameter logistic (3PL) model, which is insufficient for accurate estimation; (3) A four-round scheme generally achieves the highest performance. Increasing the number of rounds beyond four reduces the number of samples per round, leading to higher estimation variance in single-round accuracy. Based on these findings, we conclude setting $m = 4$ as a practical choice for optimal performance.

## F  TRAINING DATA SYNTHESIS

Our data synthesis procedure consists of two phases: filtering and annotation. We use DeepMath-103K (He et al., 2025) as the initial dataset. First, to construct a more challenging subset, we employ Qwen2.5-7B (Qwen et al., 2025) to sample 8 reasoning traces per instance ((temperature $= 0.6$, maximum length $= 2048$)) and retain only those instances for which all traces are incorrect. Second, we use DeepSeek-V3 (DeepSeek-AI et al., 2025b) to generate step-by-step reasoning annotations. The annotation prompt is presented below. We instruct DeepSeek-V3 to produce a logically complete and concise solution based on the reference solution provided in the original dataset.

---

**Step-by-Step Annotation Prompt**

Task: Generate a clear, step-by-step, and complete solution to the following problem with these requirements:
1. **Five or fewer Numbered Steps:**
    Ensure no logical jumps—every non-trivial inference must be justified.
2. **Key Explanations Included:**
    a. Briefly explain "why" for non-obvious steps (e.g., "We use X method because...").
    b. Avoid redefining terms/concepts already introduced.
3. **Full Calculations:**
    a. Show at least one intermediate step for computations (e.g., a = b + c $\rightarrow$ a = 5 + 3 = 8).
    b. For symbolic math, state the rules/theorems used (e.g., "By the chain rule...").
4. **Final Answer:**
    Mark clearly with \boxed{}.
**Original Question:**
{$question$}
**Reference Solution (for guidance only):**
{$reference\_solution$}
Begin your new solution:

---

## G  CHAIN-OF-THOUGHT PROMPT TEMPLATE

Following previous work (Liu et al., 2025a), we adopt the DeepSeek-R1 prompt template for both training and evaluation. This template contains minimal special tokens and is well-suited for

the base model. When competition is done, we first attempt to extract the text enclosed within `<answer> ... </answer>`, which is then used for subsequent answer extraction. If no match is found, we instead extract the content within the $\backslash$boxed{} tag.

---

**Chain-of-Thought Prompt**

A conversation between User and Assistant. The user asks a question, and the Assistant solves it. The assistant first thinks about the reasoning process in the mind and then provides the user with the answer.
User: {*question*}
Show your work in `<think> </think>` tags. And return the final answer in `<answer> </answer>` tags, for example `<answer> 12 </answer>`.
Assistant: Let me solve this step by step.
`<think>`
{*completion*}

---

## H  RELATIONSHIP BETWEEN PREDICTION ACCURACY AND HINTING RATE

We analyze the relationship between prediction accuracy and hinting rate using the Qwen2.5-3B checkpoint after 30 training steps, corresponding to the early rising stage of learning. For each example, we enumerate the hint length and prompt the model to complete the hinted problem. Accuracy is computed over 100 randomly sampled traces with a temperature setting of 1.

Figure 8 illustrates the accuracy–hinting rate curves for these 100 examples. In most cases, the curves exhibit an S-shaped trend: accuracy remains close to zero until a critical proportion of the solution is revealed, after which it rises rapidly to nearly 1. This pattern is highly consistent with the predictions of the three-parameter logistic (3PL) model.

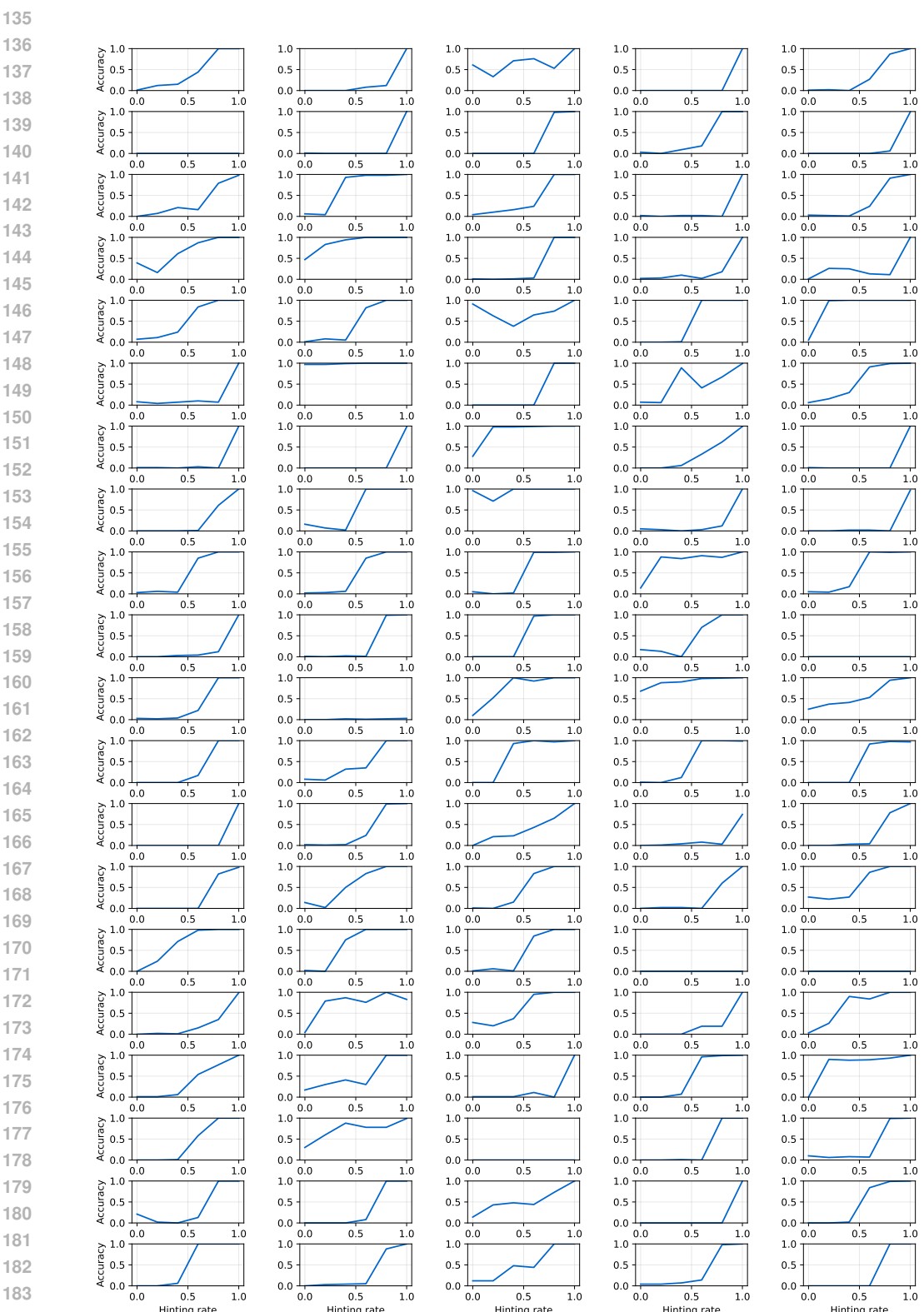

Figure 8: Accuracy with respect to the hinting rate for 100 training examples using Qwen2.5-3B.

