# OpenReview forum: "Staying in the Sweet Spot: Responsive Reasoning Evolution via Capability-Adaptive Hint Scaffolding"
_ICLR.cc/2026/Conference — Submitted to ICLR 2026_

### Official Review · Reviewer_xs8N · 2025-10-26

**Soundness:** 3
**Presentation:** 3
**Contribution:** 3
**Rating:** 6
**Confidence:** 4

**Summary:**

This paper studies how to improve training efficiency in RLVR by dynamically adjusting per-instance problem difficulty via hint scaffolding. The authors present a theoretical analysis arguing learning efficiency is upper-bounded by a quadratic function of rollout accuracy and is maximized near 50% accuracy. Building on this, they propose SEELE, a multi-round rollout scheme that fits a predictor online using an IRT/3PL model over collected (hinting rate, accuracy) pairs and appends an instance-specific hint whose length is predicted by the predictor. Experiments on multiple base models show consistent gains over GRPO, SFT, and recent hint/supervision-aided baselines.

**Strengths:**

1. The motivation is clear with theoretical analysis
2. The method is sound and novel.
3. Strong empirical gains across tasks and model families

**Weaknesses:**

1. Efficiency concerns: SEELE converts the original single-round rollout generation in GRPO into a multi-round process, which inevitably introduces additional computational overhead compared with single-round parallel generation. However, the paper does not report any efficiency metrics such as wall-clock time or overall compute cost.

2. Applicability concerns
- The multi-round rollout scheme requires a sufficient number of rollouts to fit the predictor reliably, which may limit its applicability under low-resource settings.
- The method is tightly coupled with group-based RL algorithms such as GRPO, and cannot be directly applied to algorithms like PPO that do not rely on group rollouts.
- Both the theoretical analysis and the predictor design are tailored to binary-reward settings, restricting generalization to more complex reward structures.

3. Lack of hyperparameter sensitivity analysis:
   The paper does not study how performance varies with respect to key hyperparameters such as $k_0, v_0$, leaving the robustness of the method unclear.

4. Lack of limitation discussion

5. Missing discussion of related works:
   The idea of selecting or emphasizing prompts with intermediate success rates has appeared in prior works on Prompt Curriculum and Prompt Selection [1,2,3]. However, these connections are not discussed, which would help situate SEELE within the broader landscape of adaptive prompt and curriculum methods.

[1] Self-Evolving Curriculum for LLM Reasoning\
[2] Online Difficulty Filtering for Reasoning Oriented Reinforcement Learning\
[3] Can prompt difficulty be online predicted for accelerating rl finetuning of reasoning models?

**Questions:**

see weaknesses

---

> ### Author Response · Authors · 2025-11-25
> **[W1] Effiency Concern: what is additional computational overhead of SEELE compared with single-round parallel generation?**
>
> We apologize for not presenting efficiency analysis in the paper due to the tight submission deadline. Here, we analyze the effiency from the perspective of FLOPs.
>
> Compared with GRPO under the same number of rollouts, SEELE introduces only a small number of additional FLOPs during the rollout stage because the prompt KV-cache is not shared across rounds. The extra cost from the non-linear least squares fitting is negligible, as the computation is lightweight (3 parameters) and can be parallelized across rollouts on CPU cores.
>
> To provide a quantitative comparison, we report the FLOPs of each stage in a single RL step in the table below. Following prior work [1], we approximate the FLOPs of the forward and backward passes as $2NL$ and $4NL$, respectively, where $N$ denotes the amount of LLM's non-embedding parameters and $L$ denotes the sequence length.
>
> | Method | Rollout | Old Prob. | Ref. Prob. | Update | Total |
> |---|:---:|:---:|:---:|:---:|:---:|
> | GRPO | $2NL_p + 2mn N L_r$ | $2mnN(L_p+L_r)$ | $2mnN(L_p+L_r)$ | $6mnN(L_p+L_r)$ | $2NL_p + 2mnN L_r + 10mnN(L_p+L_r)$ |
> | SEELE | $2mN L_p + 2mn N L_r$ | $2mnN(L_p+L_r)$ | $2mnN(L_p+L_r)$ | $6mnN(L_p+L_r)$ | $2mN L_p + 2mnN L_r + 10mnN(L_p+L_r)$ |
>
> Under our experimental configuration, $m=4,n=8,L_p=1024,L_r=2048$, we obtain:
> $$\frac{FLOPs(SEELE)}{FLOPs(GRPO)}=1.0055.$$
> Thus, the FLOPs increment introduced by multi-round sampling is only about $0.55\%$, which is effectively negligible. Regarding parallelism, although the generation is split into $m$ rounds, each round still produces $Bn$ trajectories, which is a sufficiently number ($B$ is the batch size, $Bn$=2048 in our experiments) to fully utilize parallel computation.
>
> We will add this analysis to the final paper.
>
> [1] Scaling Laws for Neural Language Models

---

> > ### Author Response · Authors · 2025-11-25
> > **[W3] Lack of hyperparameter sensitivity analysis for $k_0$ and $\nu_0$.**
> >
> > $k_0$ and $\nu_0$ are introduced in Line 286-297 to denote the discrimination coefficient and difficulty bias in original 3PL model. They serve only as intermediate variables that connect the original 3PL formulation to our prediction model $f_\phi$. **They are not hyperparameters and are never instantiated in our algorithm.** In the formulation presented in Line286-297, $k_0$ and $\nu_0$ are absorted in to parameters $k,\mu$.
> >
> > In our proposed prediction model $f_\phi$, all parameters $\phi=\{k,\mu,\nu\}$ are learned online through non-linear least squares. **$f_\phi$ does not contain hyperparameters requring manual assignment**. Details about model fitting are provided in Line 304-316. As illustrated in Figure 3, our prediction model $f_\phi$ can perfectly model the relationship between hint length and rollout accuracy. In Appendix H, we visualize the hint-accuracy curves for 100 exmaples, demonstrating the reasonableness of the design of $f_\phi$.

---

> ### Author Response · Authors · 2025-11-25
> **[W2] Applicability Concerns**
>
> Thank the reviewer for the thoughtful comments.
>
> ### **[W2.1] The multi-round rollout scheme requires a sufficient number of rollouts to fit the predictor reliably, which may limit its applicability under low-resource settings.**
> Although SEELE benefits from larger rollout sizes, its performance does not degrade substantially when using smaller rollout sizes ($mn=16$ and $24$), as reported in Table 3 of Appendix E. Since $f_\phi$ only contains 3 free parameters and we initialize the boundary defaults as priors (explained in Algorithm of Appendix B), its reliability is not sensitive to the number of rollouts.
>
> To further address the reviewer’s concern regarding SEELE’s effectiveness in low-resource settings, we extend our analysis to rollout sizes $mn=8$ (where $8$ is the standard group size used in GRPO) and compare SEELE with GRPO on Qwen2.5-1.5B.
>
> |Method|Rollout number ($m$x$n$)|GMS8K|MATH500|minerva|olympiadbench|aime24 (mean@32)|amc23 (mean@32)|Math Avg|ARC-c|GPQA|MMLU-Pro|General Avg|
> |---|---|---|---|---|---|---|---|---|---|---|---|---|
> |GRPO|8|26.1|11.6|5.9|6.2|0.0|8.0|9.6|62.3|**26.8**|20.5|36.5|
> |SEELE|8 (2x4)|**76.0**|**50.8**|**16.2**|**18.2**|**3.0**|**28.1**|**32.1**|**65.2**|26.3|**28.1**|**39.9**|
> |GRPO|16|67.6|35.0|9.2|9.8|2.2|16.4|23.4|66.0|24.2|22.8|37.7|
> |SEELE|16 (4x4)|**76.8**|**51.0**|**15.4**|**19.0**|**3.3**|**28.4**|**32.3**|**66.1**|**24.7**|**29.3**|**40.0**|
> |GRPO|32|70.1|36.4|10.7|11.1|1.8|15.2|24.2|64.8|25.3|23.1|37.7|
> |SEELE|32 (4x8)|**76.5**|**58.0**|**16.2**|**19.9**|**4.1**|**30.4**|**34.2**|**68.3**|**27.8**|**31.7**|**42.6**|
>
> As shown in the table above, reducing the rollout size to 8 or 16 leads to only a modest performance drop for SEELE (approximately 2% on average). In contrast, becuase we use a relative challenging training set, standard GRPO experiences substantial degradation, as smaller group sizes reduce the likelihood of sampling a correct trajectory and impair training stability. These results demonstrate that SEELE maintains a clear and consistent advantage even in low-resource scenarios.
>
>
> ### **[W2.2&W2.3] The method is tightly coupled with group-based RL algorithms and binary reward.**
> This paper focuses on the standard Reinforcement Learning with Verifiable Rewards (RLVR) setting, where rewards are typically binary and group-based RL algorithms such as GRPO, DAPO, and GSPO currently play a central role in RLVR research. Extending SEELE to arbitrary reward and value-model–based algorithms such as PPO is a promising direction, which we leave for future work.

---

> ### Author Response · Authors · 2025-11-25
> **[W4] Lack of limitation discussion**
>
> Thank you for your help in improving our paper. We will add a limitation section in the camera-ready version.

---

> ### Author Response · Authors · 2025-11-25
> **[W5] Missing discussion from pripr works on prompt curriculum and prompt selection**
>
> We thank the reviewer for raising this insightful discussion. We clarify the innovation and advantage as follows.
>
> * **Technical difference**: Literature [1,2,3] all belong to data selection. They identify “proper’’ training samples or rollouts according to their selection criteria and filter out the unsuitable ones. However, they do not modify the original problem or prompt, and therefore cannot manipulate the difficulty of each instance as SEELE does. In contrast, SEELE leverages every training sample through online, difficulty-aware hinting.
>
> * **Over concentration**: Data selection methods [1,2,3] are susceptible to over-concentration, where the algorithm repeatedly selects a narrow subset of “desirable’’ data when the model’s capability does not improve sufficiently from that subset. SEELE uniformly samples all data points and, by design, avoids this pathological concentration.
> * **Data utilization**: In data selection approaches, unselected problems make no contribution to training, limiting data utilization efficiency. SEELE, by contrast, uses every prompt by adaptively modifying its difficulty via hinting, ensuring that the model learns effectively regardless of the problem’s intrinsic difficulty. Consequently, SEELE achieves a higher performance ceiling than prompt-selection methods due to its significantly higher data utilization rate.
>
> * **Computational efficiency**: Data selection methods [1,2,3] typically require pre-computing certain metrics to assess data quality, incurring fixed computational cost even for data that are ultimately discarded. Moreover, rollout-filtering methods such as [2] require generating rollouts that are later unused for updating the model, wasting the expensive rollout budget. SEELE avoids this inefficiency by ensuring that all generated rollouts contribute to training.
>
>
> [1] Self-Evolving Curriculum for LLM Reasoning
>
> [2] Online Difficulty Filtering for Reasoning Oriented Reinforcement Learning
>
> [3] Can prompt difficulty be online predicted for accelerating rl finetuning of reasoning models?

---

### Official Review · Reviewer_3tXM · 2025-10-28

**Soundness:** 3
**Presentation:** 3
**Contribution:** 2
**Rating:** 6
**Confidence:** 3

**Summary:**

The paper introduces SEELE, a supervision-aided RLVR framework that dynamically adjusts problem difficulty. SEELE appends instance-specific partial solutions as hints and adaptively adjusts their length to maintain rollout accuracy near 50%, a theoretically justified optimum for learning efficiency. The framework employs multi-round estimation and an prediction model to infer the relationship between hint ratio and accuracy based on Item Response Theory. Experimental results show that SEELE achieves stronger generalization across multiple models and benchmarks, outperforming SFT, GRPO, and other baselines.

**Strengths:**

1. The paper is well-written and presents a coherent motivation for the proposed approach.
2. The choice of maintaining a 50% rollout accuracy is theoretically supported, and the integration of Item Response Theory provides a rigorous and interpretable framework for modeling task difficulty and hint rate.
3. SEELE achieves consistent performance gains across multiple model families and reasoning benchmarks. The paper also conducts thorough ablation studies on target difficulty levels and multi-round configurations, providing a convincing evaluation of the framework’s design efficacy.

**Weaknesses:**

1. Some notations are confusing
- The paper does not clearly specify whether $f_{\phi}$ is a global predictor shared across all instances or a local model optimized separately for each instance.
- $w$ is not defined in Algorithm 1.
2. I think the current evaluation setup does not clearly demonstrate the benefits of scaffolding compared with the original GRPO method, as the training datasets are biased toward harder examples. It would be more informative to analyze SEELE’s effectiveness across different difficulty regimes—for example, by training on (1) hard questions with hints, (2) tractable questions that the model can already solve (with or without hints), and (3) a mixed setting: training on datasets with both hard questions and tractable questions. Such a comparison would better clarify the advantages of adaptive scaffolding methods, i.e., learning from harder examples.

**Questions:**

1. Could you clarify the "j" in Figure 2?
2. For the partial solutions used as hints, were only correct reasoning traces included, or were all generated traces (including incorrect ones) used? Additionally, is this the same dataset used in SFT, and how many reasoning traces per question were retained?
3. What is the computational overhead introduced by the multi-round estimation compared with original GRPO? It would be helpful to include a quantitative comparison showing total wall-clock cost.

---

> ### Author Response · Authors · 2025-11-25
> **[W1&Q1] Notation Clarification**
>
> We apologize for the unclear notation. Due to the tight submission timeline, the current manuscript is not fully polished. We provide detailed clarifications below and will carefully refine the notation in the final camera-ready version.
> ### **[W1.1] Whether $f_\phi$ is a global predictor shared across all instances or a local model optimized separately for each instance.**
> $f_\phi$ is a local model optimized separately for each instance.
>
> ### **[W2.2] $w$ is not defined in Algorithm 1.**
> In Algorithm 1 Line 10, $w$ is a dummy variable defined within the set $\\{(w,w)|w\in\\{0,1\\}\backslash\\{p_x^{(j)}\\}_{j=1}^{i-1}\\}$.
>
> This set represents the boundary value (0 and 1) that $p_x$ has not covered in the previous $i-1$ rounds. We assigns the default accuracy (0 for $p_x=0$ and 1 for $p_x=1$) for the unseen boundary to stablize the modeling of $f_\phi$.
>
> ### **[Q1] Could you clarify the "j" in Figure 2?**
> In Figure 2, $\\{(p^{(j)},a^{(j)})\\}$ is the short for $\\{(p^{(j)},a^{(j)})\\}_{j=1}^{ni}$, which represents all hint-accuracy pairs sampled in past $i$ rounds. Here $j$ refers the index for $j$-th stored pair.

---

> ### Author Response · Authors · 2025-11-25
> **[W2] It would be more informative to analyze SEELE’s effectiveness across different difficulty regimes.**
>
> We thank the reviewer for suggesting this perspective of analysis. We follow the reviewer's requirements and conduct such an experiment on Qwen2.5-1.5B (due to limited rebuttal time, we currently cannot test larger models). We categorize a problem as easy if the model achieves a positive pass@16 score on it. Conversely, we assign questions with pass@16 = 0 to the hard set. Now, we have split the original training set into easy and hard partitions. The original set represents the mixed setting.
>
> |Difficulty|Method|GMS8K|MATH500|minerva|olympiadbench|aime24 (mean@32)|amc23 (mean@32)|Math Avg|ARC-c|GPQA|MMLU-Pro|General Avg|
> |---|---|---|---|---|---|---|---|---|---|---|---|---|
> |Easy|GRPO|67.2|31.0|11.4|9.8|1.0|15.3|22.6|**64.5**|20.7|23.1|36.1|
> |Easy|SEELE|**72.4**|**46.6**|**13.2**|**16.6**|**2.6**|**19.6**|**28.5**|64.1|**23.2**|**24.8**|**37.4**|
> |Hard|GRPO|71.3|40.0|10.7|11.6|2.3|15.9|25.3|65.2|**25.8**|27.7|39.6|
> |Hard|SEELE|**75.7**|**58.4**|**16.5**|**20.4**|**4.3**|**30.4**|**34.3**|**67.9**|**25.8**|**31.2**|**41.6**|
> |Mixed|GRPO|70.1|36.4|10.7|11.1|1.8|15.2|24.2|64.8|25.3|23.1|37.7|
> |Mixed|SEELE|**76.5**|**58.0**|**16.2**|**19.9**|**4.1**|**30.4**|**34.2**|**68.3**|**27.8**|**31.7**|**42.6**|
>
> As shown in the table above, across all difficulty settings, SEELE consistently outperforms GRPO significantly, demonstrating its general effectiveness. We will add these results to the appendix in the camera-ready version.

---

> ### Author Response · Authors · 2025-11-25
> **[Q2] (1) Were only correct reasoning traces included, or were all generated traces (including incorrect ones) used; (2) Is the hint dataset identical to the dataset used in SFT; (3) How many reasoning traces per question were retained?**
>
> We apologize for not presenting these details clearly in the paper. Below are the clarification and will be added in the final paper.
>
> 1. We call DeepSeek-V3 to generate complete solutions for each problem and only keep the correct ones matching the labeled answers. More details about hint annotation are included in Appendix F.
> 2. Hint dataset is exactly the same dataset used in SFT. SFT directly trains the model to imitate the hints.
> 3. For each problem, we retain 1 reasoning trace as the full hint. During training, we dynamically concatenate part of this trace behind the prompt.

---

> ### Author Response · Authors · 2025-11-25
> **[Q3] What is the computational overhead introduced by the multi-round estimation compared with original GRPO?**
>
> We thank the reviewer for suggesting that we include an analysis of the computational cost. We apologize for not presenting efficiency analysis in the paper due to the tight submission deadline. Here, we analyze the effiency from the perspective of FLOPs.
>
> Compared with GRPO under the same number of rollouts, SEELE introduces only a small number of additional FLOPs during the rollout stage because the prompt KV-cache is not shared across rounds. The extra cost from the non-linear least squares fitting is negligible, as the computation is lightweight (3 parameters) and can be parallelized across rollouts on CPU cores.
>
> To provide a quantitative comparison, we report the FLOPs of each stage in a single RL step in the table below. Following prior work [1], we approximate the FLOPs of the forward and backward passes as $2NL$ and $4NL$, respectively, where $N$ denotes the amount of LLM's non-embedding parameters and $L$ denotes the sequence length.
>
> | Method | Rollout | Old Prob. | Ref. Prob. | Update | Total |
> |---|:---:|:---:|:---:|:---:|:---:|
> | GRPO | $2NL_p + 2mn N L_r$ | $2mnN(L_p+L_r)$ | $2mnN(L_p+L_r)$ | $6mnN(L_p+L_r)$ | $2NL_p + 2mnN L_r + 10mnN(L_p+L_r)$ |
> | SEELE | $2mN L_p + 2mn N L_r$ | $2mnN(L_p+L_r)$ | $2mnN(L_p+L_r)$ | $6mnN(L_p+L_r)$ | $2mN L_p + 2mnN L_r + 10mnN(L_p+L_r)$ |
>
> Under our experimental configuration, $m=4,n=8,L_p=1024,L_r=2048$, we obtain:
> $$\frac{FLOPs(SEELE)}{FLOPs(GRPO)}=1.0055.$$
> Thus, the FLOPs increment introduced by multi-round sampling is only about $0.55\%$, which is effectively negligible. We will add this analysis to the final paper.
>
> [1] Scaling Laws for Neural Language Models

---

### Official Review · Reviewer_yejB · 2025-11-01

**Soundness:** 2
**Presentation:** 2
**Contribution:** 2
**Rating:** 2
**Confidence:** 5

**Summary:**

SEELE is a supervision-aided RLVR framework that fixes a key problem in current RL-with-verifiable-rewards methods: the task difficulty often doesn’t match the model’s current ability, which makes exploration inefficient. SEELE keeps training inside that region by adding hints to each problem and dynamically choosing the hint length per instance so that the task is neither too hard nor too easy. To pick that hint length, it runs multi-round rollouts and fits an item-response-theory (IRT) model on accuracy–hint pairs to predict the best hint for the next round. The authors evaluate the method over several reasoning benchmarks.

**Strengths:**

1. The paper introduces SEELE, a method that leverages instance-level hints to make RLVR exploration more effective.
2. The method is evaluated on several math-reasoning benchmarks and shows consistent gains even with relatively small models.

**Weaknesses:**

1. The paper should clarify how the hints are produced and whether they risk leaking too much target information, effectively turning the setup into SFT; more analysis of how different hint lengths affect training dynamics would help.
2. In Figure 4, reward increases while response length drops sharply and accuracy improves only modestly, which suggests the training dynamics may not be fully understood or may be unstable.
3. The experiments are limited to small models (e.g., Qwen2.5-3B); it would be stronger to show results on larger models (e.g., Qwen2.5-32B or QwQ-32B) to confirm the method scales and is not just fixing small-model artifacts.

**Questions:**

Please refer the limitations.

---

> ### Author Response · Authors · 2025-11-25
> **[W1] (1) How the hints are produced; (2) Whether hints turn the setup into SFT; (3) How different hint lengths affect training?**
>
> Thank you for your question regarding the hinting mechanism.
> 1. **How the hints are produced**: As stated in Line324-326 in the paper, we use DeepSeek-V3 to annotate step-by-step reasoning traces for each training problem, which serve as the full hints. The annotation prompt used is provided in Appendix E. During training, SEELE appends a prefix of the annotated trace behind the original prompt as the hint.
> 2. **Whether hints turn the setup into SFT**: In Table 1, we have reported the SFT results with the hints as the target. Its performance is substantially lower than SEELE, demonstrating that SEELE’s improvements are not due to collapsing into standard SFT.
> 3. **How different hint lengths affect training**: To investigate this, we conduct an analytical experiment on Qwen2.5-1.5B and control the hint length from zero to full trace (length is fixed during training). Due to the limited rebuttal time, we cannot evaluate more models, but we expect the results to generalize.
>
> |Hint length|GMS8K|MATH500|minerva|olympiadbench|aime24 (mean@32)|amc23 (mean@32)|Math Avg|ARC-c|GPQA|MMLU-Pro|General Avg|
> |---|---|---|---|---|---|---|---|---|---|---|---|
> |0 (GRPO)|70.1|36.4|10.7|11.1|1.8|15.2|24.2|64.8|25.3|23.1|37.7|
> |1|69.8|35.8|8.5|10.8|1.9|17.3|24.0|66.5|25.8|23.8|38.7|
> |2|74.0|53.2|14.0|17.2|2.9|27.6|31.5|65.4|27.8|28.8|40.7|
> |3|74.2|53.0|15.8|18.1|2.7|26.6|31.7|66.4|25.8|29.3|40.5|
> |4|76.3|50.2|14.0|19.3|2.5|26.9|31.5|66.7|19.2|30.0|38.6|
> |Full (SFT)|67.4|43.6|13.6|12.6|1.4|16.4|25.8|63.7|25.8|30.2|39.9|
> |SEELE|**76.5**|**58.0**|**16.2**|**19.9**|**4.1**|**30.4**|**34.2**|**68.3**|**27.8**|**31.7**|**42.6**|
>
> The results suggest that SEELE's hinting length managing strategy generally outperforms exclusive SFT or GRPO. Providing more hints does not necessarily lead to better performance. Crucially, it is essential to identify the most appropriate length, which is precisely the objective of our approache SEELE.
> We will add these results in the camera-ready verion.

---

> ### Author Response · Authors · 2025-11-25
> **[W2] Whether the response length drop in early stage indicates a training instability？**
>
> Thank you for pointing out the early-stage response-length behavior.
>
> **The answer is No.** We follow the DeepSeek-R1-Zero setting and train directly from a base model. Under this setup, an early-stage drop in response length is a common phenomenon that has been consistently reported in prior works [1, 2, 3, 4]. Existing studies also show that this behavior does not negatively affect subsequent training. As demonstrated in Figure 4, the response length increases steadily after reaching its minimum point.
>
> [1] Efficient RL Training for Reasoning Models via Length-Aware Optimization
>
> [2] ProRL: Prolonged Reinforcement Learning Expands Reasoning Boundaries in Large Language Models
>
> [3] DAPO: An Open-Source LLM Reinforcement Learning System at Scale
>
> [4] Learning to Reason under Off-Policy Guidance

---

> ### Author Response · Authors · 2025-11-25
> **[W3] The experiments are limited to small models (Size < 32B)**
>
> Thank you for raising the concern regarding model scale.
>
> As RL training is time-consuming, for a 7B model, we have to spend 2-3 weeks to test our approach and other baselines. For larger models such as 32B, due the limited rebuttal time and computational resources, we currently can not run experiments, but we expect SEELE’s benefits to generalize given the trends observed on smaller models.
>
> We have conducted experiments on models with 1.5B, 3B, and 7B parameters. SEELE exhibits consistent improvements over the baselines across all model sizes.

---

### Official Review · Reviewer_Knso · 2025-11-02

**Soundness:** 3
**Presentation:** 2
**Contribution:** 2
**Rating:** 2
**Confidence:** 4

**Summary:**

The paper presents Seele, a multi-round hint framework, for RL-Finetuning of LLMs, which is adaptable to problem difficulty. The authors begin by providing a theoretical foundation for identifying how rollout accuracy correlates with learning efficiency. They then propose an adaptive multi-round framwork

**Strengths:**

- Theoretical contribution for relationship between learning efficiency and learning accuracy
- Empirical evaluation on a wide range of math + general domain reasoning benchmarks.

**Weaknesses:**

- Measuring the difficulty from the average success rate + hinting has been studied in prior work, making the contribution strictly the integration of these approaches, which seems limited from a novelty perspective.
- Using all mxn rollouts for advantage calculation seems incorrect, in particular for the baseline computation, where the hint is different per round. A value baseline (V(s)) should strictly be a function of the state, which isn't static in this instance, given that the loss is computed only on generated tokens.
- The design decisions (e.g 3PL and multi-round sampling) aren't independently ablated, making it unclear what the contribution of each component is to the approach

**Questions:**

- How does the flop equivalent performance compare for Seele vs other approaches due to additional adaptable component contributing to additional computation?
- How does Seele work for LongCOT models (responses are quite short at 800 tokens) with lower scores on math reasoning benchmarks?
- See Weaknesses above.

---

> ### Author Response · Authors · 2025-11-25
> **[W1] What is SEELE's novelty compared with prior works using hinting?**
>
> Thank you for your thoughtful comments.
> * **Technical novelty**: Prior hinting methods [1,2,3] are static, where hints are applied and indiscriminately across all training problems, regardless of problem difficutly. Additionally, the hinting level are pre-scheduled before training, without considering the changing model capability during training. **Our approach provides an online feedback mechanism to dynamically modulates the hint level towards the optimal learnability. SEELE not only measures difficulty but also modifies it on-the-fly during rollout—an ability that, to our knowledge, has not been explored in any previous work.**
> * **Theoretical novelty**: Prior works [1,2,3] do not provide a principled formulation of how hinting should relate to learning efficiency. **Our paper offers a new theoretical explanation characterizing the relationship between problem difficulty and learning efficiency, which directly guides us to adapt hint levels toward a target 50% rollout accuracy.**
>
> [1] UFT: Unifying Supervised and Reinforcement Fine-Tuning.
>
> [2] Blending Supervised and Reinforcement Fine-Tuning with Prefix Sampling.
>
> [3] Learning to Reason under Off-Policy Guidance.

---

> ### Author Response · Authors · 2025-11-25
> **[W2] Why computing advantage function using all $m$x$n$ rollouts with different hints is valid for GRPO-like algorithms, given that the baseline is not a static function of the state $s$?**
>
> We thank the reviewer for this insightful question. Below we clarify why our design is fully compatible with the theoretical requirements of policy gradient methods and how it relates to the original GRPO formulation.
>
> ## **Validity as a baseline**
>
> In policy gradient methods, subtracting any baseline that is **independent of the sampled action** does not introduce bias:
>
> $$
> E_{\pi_{\theta}(\cdot|s)}[\nabla_{\theta} \log\pi_{\theta}(a|s)\, b(s)] = \
> b(s)E_{a\sim\pi_{\theta}(\cdot|s)}[\nabla_{\theta} \log\pi_{\theta}(a|s)] = 0.
> $$
>
> **While hints slightly shift the induced state distribution, they do not affect the validity of using a scalar baseline**. The policy gradient theorem imposes no requirement that the baseline must approximate $V(s)$; it only requires action-independence.
>
> ## **Relationship with GRPO**
>
> GRPO’s intra-group normalization was introduced for convenience when prompts are identical. In our setting, *adaptive hinting acts as a curriculum mechanism*, improving the signal quality of rollouts, while the global baseline provides a statistically more reliable centering term. Together they form a principled variance-reduction mechanism that preserves unbiasedness.

---

> ### Author Response · Authors · 2025-11-25
> **[W3] Multi-round sampling and 3PL model are not independently abalated.**
>
> We thank the reviewer for raising this ablation question.
>
> First, we would like to clarify the relationship between multi-round sampling and 3PL model. 3PL model is adopted to manage the hint length manipulation during multi-round sampling. So, **multi-round sampling cannot be independently ablated from 3PL model**, becuase 3PL has no effect in standard single-round sampling.
>
> **Next, following the reviewer’s request, we provide a 3PL ablation.** We construct a naive control strategy that assigns random hint lengths to each round, thereby removing the 3PL mechanism. Due to limited rebuttal time and computational resource, we currently conduct ablation study only on Qwen2.5-1.5B. The results are shown in the following table:
>
> |Length Control|GMS8K|MATH500|minerva|olympiad|aime24 (mean@32)|amc23 (mean@32)|Math Avg|ARC-c|GPQA|MMLU-Pro|General. Avg|
> |---|---|---|---|---|---|---|---|---|---|---|---|
> |Random|74.3|53.2|13.6|19.1|2.5|29.1|32.0|64.6|19.2|30.4|38.1|
> |3PL|**76.5**|**58.0**|**16.2**|**19.9**|**4.1**|**30.4**|**34.2**|**68.3**|**27.8**|**31.7**|**42.6**|
>
> Without 3PL model, SEELE's performance drop 2.2\% and 4.5\% on average on Math and General benchmarks, respective, demonstrating its importance.
> We will add these results to the final version of the paper.

---

> ### Author Response · Authors · 2025-11-25
> **[Q1] FLOPs Comparision between SEELE and other approaches.**
>
> We thank the reviewer for suggesting that we include an analysis of the computational cost.
>
> In short, SEELE has nearly identical FLOPs to other RL methods (under the same number of rollouts). The only additional cost over GRPO comes from multi-round sampling because the prompt KV-cache is not shared across rounds. The extra cost from the non-linear least squares fitting is negligible, as the computation is lightweight (3 parameters) and can be parallelized across rollouts on CPU cores.
>
> To provide a quantitative comparison, we report the FLOPs of each stage in a single RL step in the table below. Following the prior work [1], we approximate the FLOPs of the forward and backward passes as $2NL$ and $4NL$, respectively, where $N$ denotes the amount of LLM's non-embedding parameters and $L$ denotes the sequence length.
>
> | Method | Rollout | Old Prob. | Ref. Prob. | Update | Total |
> |---|:---:|:---:|:---:|:---:|:---:|
> | GRPO | $2NL_p + 2mn N L_r$ | $2mnN(L_p+L_r)$ | $2mnN(L_p+L_r)$ | $6mnN(L_p+L_r)$ | $2NL_p + 2mnN L_r + 10mnN(L_p+L_r)$ |
> | SEELE | $2mN L_p + 2mn N L_r$ | $2mnN(L_p+L_r)$ | $2mnN(L_p+L_r)$ | $6mnN(L_p+L_r)$ | $2mN L_p + 2mnN L_r + 10mnN(L_p+L_r)$ |
>
> Under our experimental configuration, $m=4,n=8,L_p=1024,L_r=2048$, we obtain:
> $$\frac{FLOPs(SEELE)}{FLOPs(GRPO)}=1.0055.$$
> Thus, the FLOPs increment introduced by multi-round sampling is only about $0.55$%, which is effectively negligible.
> We will include this analysis in the final paper.
>
> [1] Scaling Laws for Neural Language Models.

---

> ### Author Response · Authors · 2025-11-25
> **[Q2] How does Seele work for LongCOT models?**
>
> Thank you for your advice.
> We follow the reviewer's requirement to verfiy SEELE's effectiveness on LongCoT models. We select the long-CoT model *DeepScaleR-1.5B-Preview* as the base model and train it with SEELE and standard GRPO, respectively.
>
> |Method|GMS8K|MATH500|minerva|olympiad|aime24 (mean@32)|amc23 (mean@32)|Math Avg|ARC-c|GPQA|MMLU-Pro|General. Avg|
> |---|---|---|---|---|---|---|---|---|---|---|---|
> |GRPO|69.4|59.0|8.5|40.8|21.5|57.2|42.7|**8.9**|18.7|14.8|14.1|
> |**SEELE**|**73.8**|**63.6**|**18.4**|**41.8**|**21.7**|**58.4**|**46.3**|6.2|**29.8**|**19.9**|**18.6**|
>
> As shown in the table above SEELE outperforms GRPO by 3.6 points avergely on math benchmarks, and 4.5 points on general domain tasks on average. This significant improvment demonstrate that SEELE can generalize to LongCoT training.
>
> We will add these results in the camera-ready verion.

---

### Author Response · Authors · 2025-11-27
**Further Discussion**

We sincerely thank the AC and all reviewers for their time and effort in evaluating our paper. We summarize the reviewers’ comments and our responses as follows.

### **Strengths**
* **Theoretical Contribution**: The design of SEELE is theorectically supported by deriving the relationship between learning efficiency and learning accuracy (noted by Knso, 3tXM, and xs8N).
* **Novelty**: the method is sound and novel (noted by xs8N).
* **Sound Empirical Evalution**: SEELE achieves consistent performance gains across multiple model families and reasoning benchmarks. (noted by Knso, yejB, 3tXM, and xs8N)

### **Weaknesses and Rebuttal**
* **Efficiency Concerns**: The reviewers are concerned about whether SEELE incurs substantial additional computational cost (noted by Knso, 3tXM, and xs8N). *To address this concern, we provide a detailed FLOPs analysis and explained that SEELE's computation overhead is nearly identical to the standard GRPO, introducing only 0.55\% additional overhead.*
* **Missing Discussion of Related Works**: The reviewers raise a concern that the idea of selecting prompts with intermediate success rates and hinting have appeared in prior works. *We include a comprehensive discussion clarifying that our online multi-round sampling and feedback framework is novel and provides capabilities not present in prior work.*
* **Method Design Concerns**: The question the correctness of the advantage calculation in the multi-round sampling and the effectiveness of each components (noted by Knso). *To address the concerns, we present (1) a math analysis showing our advantage calculation does not violate the requirements of policy gradient, and (2) a direct ablation study to demonstrate the effectiveness of each components.*
* **Extended Experimental Setting**: The reviewers question SEELE's performance on LongCoT models (Knso), larger models (yejB), and in hint-length-control (yejB), and difficulty-control settings (3tXM). *We provide all requested experiments except scaling to very large models due to limited rebuttal time and computational resources. The results confirm the robustness and effectiveness of SEELE across diverse model types and training configurations.*
* **Applicability concerns**: reviewer xs8N is concerned about SEELE's applicability in low-rollout settings. *We conduct expriments with a standard GRPO rollout size, demonstrating SEELE's prominent advantage even with constrained rollout budgets.*
* **Unclear notation, Hyperparameter, and Hinting Details**: reviewer 3tXM and xs8N are confused with some notations (e.g. $k_0$, $v_0$). *We have clarified the notation's meaning and whether they are hyperparameters, and provided detailed explanations for all hint-related components.*

We again thank the AC and all reviewers for reading our rebuttal. We will incorporate all the additional discussions and experiments into the final version of the paper. We are ready to discuss any further question about our paper.

---

### Meta-Review · Area_Chair_LQes · 2026-01-07

**Summary:**

This paper proposes SEELE, a supervision-aided RLVR/RL-with-verifiable-rewards framework that appends partial solution “hints” dynamically to each training instance to keep problem difficulty controlled (targeting ~50% rollout accuracy). The method uses multi-round rollouts and fits an IRT/3PL-style predictor online from (hint-length, accuracy) pairs to choose subsequent hint lengths. The experiments report gains over GRPO, SFT, and prior supervision-aided approaches on several math and general reasoning benchmarks.

Reviewers were split but overall leaned negative, with two confident rejects (both rating 2) and two borderline positives (both rating 6). The main concerns driving the recommended reject are: (i) unclear novelty relative to prior work on hinting, prompt curriculum/selection, and supervision-aided RLVR; (ii) methodological clarity/correctness questions around the multi-round, hint-modified rollouts and how advantage/baseline computation aligns with GRPO-style objectives; and (iii) evaluation and applicability gaps, especially around fairness/interpretability of comparisons, practical efficiency (wall-clock vs FLOPs), and limited evidence for scaling/generalization beyond small-to-mid sized models. While the rebuttal added analyses and extra experiments, the remaining uncertainty about conceptual novelty and evaluation rigor is substantial enough that the paper does not clear the acceptance bar.

**Reviewer Concerns:**

Concerns fully or mostly addressed by the rebuttal:
* Efficiency overhead (partial): Authors provide a FLOPs-based accounting and argue minimal overhead vs GRPO under matched rollouts, and discuss parallelism assumptions. This helps, though it does not fully substitute for wall-clock measurements.
* LongCoT applicability: Added results on a LongCoT model showing consistent gains over GRPO.
* Hint mechanism details / “turning into SFT”: Authors clarify hint generation (external model-generated traces; keep only correct), confirm SFT uses the same hint dataset, and provide comparisons suggesting SEELE > SFT-with-hints.
* Some ablation/sensitivity requests: Added a 3PL-vs-random-length control ablation (limited to one model), hint-length sweep, rollout-budget experiments, and a difficulty-regime split (easy/hard/mixed) showing consistent gains.

Concerns still outstanding / only partially addressed
* Novelty vs related work (outstanding): While authors argue “online instance-level hint-length control” is new, the rebutal does not fully establish a clear separation from prior curriculum/prompt-selection/hinting approaches via strong, directly comparable baselines or sharper positioning. The contribution still reads to some reviewers as an integration of known ideas.
* Correctness/appropriateness of the learning objective and estimator (partially addressed): The rebuttal argues the baseline is action-independent and thus unbiased, but concerns remain about whether aggregating rollouts across different hinted prompts yields a clean comparison to GRPO-like objectives and whether the estimator is well-justified beyond unbiasedness.
* Evaluation rigor / fairness and interpretability (outstanding):
   * Practical efficiency remains unclear without wall-clock/runtime and systems-level reporting; FLOPs alone may not reflect multi-round rollout costs and KV-cache behavior.
   * The reliance on external, correct-only traces raises questions about general applicability and how much of the improvement is attributable to high-quality supervised signal rather than the proposed adaptive RL mechanism.
   * Scaling evidence remains limited (up to 7B; no large-model confirmation), which is a significant gap for a training-framework contribution. (not a deal-breaker for me considering limitations in compute cost and infrastructure)

**Reviewer Scores:**

* Reviewer Knso: Unlikely to change, or increase to 4 at most. Rationale: Some concerns were addressed (FLOPs analysis; LongCoT results; a 3PL-related ablation), but the reviewer’s core objections on novelty and the baseline/advantage validity under changing hints are only partially resolved.
* Reviewer yejB: Unlikely to change, or increase to 4 at most. Rationale: Authors clarified hint generation and provided hint-length experiments and discussion of length dynamics. However, the reviewer’s skepticism about training dynamics understanding and the absence of strong scaling evidence likely keeps them below threshold.
* Reviewer 3tXM:  Possible change 6 to 8. Rationale: Weaknesses were not crucial. Notation questions were answered; compute overhead addressed (again via FLOPs); and the requested analysis across difficulty regimes was provided.
* Reviewer xs8N: Likely to maintain score (6). Rationale: Low-rollout experiments and additional discussion of connections to curriculum/prompt selection likely increase confidence, but limitations (binary reward / group-based coupling; efficiency still mostly FLOPs) may bound enthusiasm.

---

### Decision · Program_Chairs · 2026-01-26

Reject